# DNA diamond formulates a decomposable composite letter constellation model for DNA data storage

Qi Ge [1], Menghui Ren [1], Tingting Qi [1], Changcai Han [1], Yingjin Yuan [2,3] ✉ & Weigang Chen [1,2,3] ✉

Oligonucleotide multiplicity is an inherent property of current DNA synthesis technology. Composite letter DNA storage exploits this property to improve logical density and reduce costs. However, letter indistinguishability and high molecular diversity pose challenges for reliable recovery. Here, we formulate a composite letter constellation model, named DNA diamond, consisting of 15 decomposable points. Inspired by set partitioning in telecommunications, we propose a two-stage letter detection framework that partitions these letters into four distinguishable subsets based on their discrete entropy. Furthermore, we incorporate encoded double-end indices to eliminate crosstalk between synthesis sites and simultaneously apply length filtering to suppress error propagation during readout. We validate the eight-letter and 15-letter composite letter DNA storage under DNA diamond model, each with 10,000 composite strands. The eight-letter system achieves a payload density of 2.5 bits per letter and enables error-free recovery at 14× coverage, surpassing the storage density of prior six-letter systems while requiring lower coverage. The full 15-letter constellation enables 3.125 bits per letter for payload with error-free recovery at 33× coverage, corresponding to a density of 2.23 bits per letter for payload plus indices. The proposed decomposable DNA diamond model advances a practical and scalable framework for high-density composite DNA data storage.

With the explosive growth of global data, traditional storage media may not satisfy the demands of future long-term data storage[1–4]. DNA data storage, which encodes digital information into synthetic DNA, has emerged as a promising alternative[5–8]. The theoretical upper limit of 2 bits per synthesis cycle (per nucleotide) is established by using four natural DNA bases (e.g., A↔00, T↔01, G↔10, C↔11). Recent advancements have demonstrated an achievable information density of up to 1.57 bits per nucleotide using only natural bases[9]. However, due to the limited number of natural bases, it remains challenging to further increase information density in a cost-efficient manner[10–13]. To address this limitation and reduce overall costs, two approaches have been explored: (i) incorporating non-natural bases to expand the base variety[14–17] and (ii) employing composite letters[18,19]. Composite letters, which are mixtures of the four natural bases in controllable ratios, effectively enhance logical density (Fig. 1a). Given that multiple copies of an individual DNA sequence are synthesized at a unique synthesis site (Fig. 1b), composite letters exploit the inherent redundancy of DNA synthesis[20–23].

Despite these advantages, composite DNA storage faces challenges during readout, particularly elevated error rates and the

[1]School of Microelectronics, Tianjin University, Tianjin, China. [2]State Key Laboratory of Synthetic Biology, Tianjin University, Tianjin, China. [3]Frontiers Science Center for Synthetic Biology (Ministry of Education), School of Synthetic Biology and Biomanufacturing, Tianjin University, Tianjin, China. ✉e-mail: yjyuan@tju.edu.cn; chenwg@tju.edu.cn

**Fig. 1 | A decomposable constellation diamond for high-density composite letter DNA storage. a** The DNA diamond model defines 15 decomposable points on a tetrahedron formed by {A, T, G, C}, increasing theoretical density from 2.0 to 3.9 bits per letter. **b** Ordinary strands collectively form a composite strand. **c** Letter detection error increases with the number of constituent bases ($K$). Dots indicate individual measurements; bars show mean ± SD ($K = 1$, $n = 12$; $K = 2$, $n = 62$; $K = 3$, $n = 47$; $K = 4$, $n = 12$). **d** Required sequencing coverage for error-free recovery increases with alphabet size. **e** Set-partitioning detection: The entropy guides a partitioning of the constellation into four subsets to reduce the candidate letter space within each subset. Different colors denote different composite alphabets: ATGC (blue), RYMKSW (red), HBVD (green), and N (orange). **f** Sequencing coverage and logical density compared with the state-of-the-art methods. Source data are provided as a Source data file.

requirement of high sequencing coverage. These issues stem from two fundamental and interacting sources. First, synthesis and sequencing errors perturb the intended base mixture proportions. As the mixture complexity increases, this perturbation leads to higher inference error rates for composite letters at a given sequencing depth (Fig. 1c). Second, statistical sampling bias at low sequencing coverage leads to

fluctuations in the observed base-frequency distributions, making it difficult to accurately infer the original composition. Together, these factors reduce the discriminability of similar base mixtures, thereby increasing error rates and necessitating higher coverage to ensure reliable readout (Fig. 1d). For example, in a DNA storage system with a 15-letter composite alphabet[18], error-free data recovery required a

sequencing coverage of at least 250×, whereas conventional natural DNA data storage only required about 5× coverage[24,25]. The molecular diversity increases sharply as the length of the composite strand increases and thus renders the traditional clustering scheme ineffective[19]. An effective scheme is needed to identify molecules originating from the same synthesis site, as these molecules collectively represent a single composite letter. However, this aspect has not been thoroughly explored in existing studies. In prior work with a six-letter composite alphabet ($\Sigma_6$), Anavy et al.[19] demonstrated data recovery at 29× sequencing coverage using Kullback–Leibler (KL) divergence inference, Fountain coding, and Reed–Solomon (RS) coding[26–28]. More recently, Xu et al.[29] applied a soft-decision decoding algorithm (Derrick-cp) to the same alphabet, reducing the practical coverage threshold to 17×. For larger composite alphabets, such as a 256-letter scheme targeting ultra-high logical density (theoretically up to 8 bits/letter), simulations by Anavy et al. showed that approximately 2000× coverage would be required. The recent Derrick-cp algorithm reduced this requirement to 490×, highlighting the potential of advanced decoding methods[29]. Importantly, while increasing the composite alphabet size enhances storage density, it also inherently raises the required sequencing depth. Offsetting this trade-off requires robust detection and decoding strategies, such as those incorporating soft information, which can substantially lower coverage requirements and reduce overall sequencing costs. Nowadays, there are some theoretical investigations on the design of composite letter DNA storage[30–32]. In this study, we adopt set partitioning (SP), an information-theoretical paradigm widely used in communication modems, to rationally devise the composite alphabet for DNA storage.

Here, we develop an information-theoretic framework to achieve low-coverage and reliable data readout for composite letter DNA data storage (Fig. 1e). First, we formulate a decomposable constellation model, termed DNA diamond. It comprises 15 points and increases the theoretical logical density to $\log_2(15) \approx 3.9$ bits per synthesis cycle (per letter). Second, leveraging the discrete property of the information entropy of these 15 letters, we propose a two-stage composite letter detection method (Supplementary Fig. 1). These 15 letters are partitioned into four subsets based on their discrete entropy values. Maximum likelihood estimation (MLE) is then applied within each subset to infer specific composite letters. Third, given that reliable index sequences are critical for accurate data readout[33], we propose the encoded double-end indices to eliminate crosstalk. Simultaneously, we introduce a length filtering step to discard corrupted reads with indel errors. To verify the proposed scheme, we designed error correction codes tailored to different alphabets and conducted experiments on both column-based and array-based synthesis platforms, along with simulation tests (Supplementary Figs. 2–4). Specifically, we implemented the eight-letter alphabet using column-based synthesis. Error-free data recovery was achieved at 14× sequencing coverage. This scheme increased the information density from the canonical 2 bits per letter to 2.5 bits per letter (excluding primers and indices). The logical density of payload and indices was 1.97 bits per letter. In comparison with state-of-the-art schemes, our system achieved comparable performance in terms of density and sequencing coverage (Fig. 1f and Supplementary Table 1). We further designed large-scale oligonucleotide pools containing 10,000 composite strands. Both the eight-letter and 15-letter alphabets were experimentally validated using array-based synthesis. For the 15-letter system, our composite letter detection method enabled error-free recovery at 33× sequencing coverage. The system achieved a logical density of 3.125 bits per letter for payload, corresponding to a density of 2.23 bits per letter for payload and indices. Our work validated the feasibility of using a 15-letter alphabet for DNA data storage (10,000 strands synthesized in a single pool) and demonstrated that error-free recovery can be achieved at relatively low sequencing coverage.

## Results

### Decomposable DNA diamond constellation model and signal set partitioning

The composite DNA storage faces two previously unencountered challenges. First, the letter error rate increases with the number of constituent natural bases per composite letter[19]. Second, the number of unique DNA strands is determined by the number of composite letters and increases explosively as the alphabet extends (Supplementary Figs. 5 and 6). High error rates and sequence diversity impose greater demands on sequencing coverage to ensure reliable data recovery. To overcome these limitations, we developed a model for composite letter DNA storage and adopted a well-established concept from information theory to improve composite letter detection.

First, we proposed the DNA diamond, a 15-letter decomposable alphabet constellation model. The DNA diamond can be geometrically represented as a tetrahedral simplex whose four vertices represent the natural bases {A, T, G, C}. Composite letters are formed by combining these bases in defined ratios and can be represented as four-dimensional vectors. Fifteen representative points are selected on the tetrahedron to form a finite set of decomposable composite letters, including its vertices, edge centers, face centers, and the centroid. Unlike previous resolution-based composite alphabets[19], the finite set of 15 letters exhibits only four discrete information entropy values (Supplementary Fig. 7, Supplementary Table 2). This property enables entropy-guided partitioning of 15 letters into four distinguishable subsets.

Then, we developed an innovative two-stage composite letter detection framework (Supplementary Fig. 8). In the first stage, entropy-based SP separates the 15 letters into four subsets. In the second stage, MLE is employed to infer the optimal result. In practice, synthesis and sequencing noise broaden the originally discrete entropy values into distinguishable distributions, as observed in experimental data. This strategy, adapted from coded modulation techniques in telecommunications[34,35], reduces the error rate in composite letter detection. A similar entropy-based method has been applied in the base calling of Solexa sequencing[36]. Building on this composite letter detection approach, we established a complete data recovery pipeline. In brief, it begins with paired-end sequencing reads and involves primer identification, length filtering, index identification, and composite letter detection within each group (Supplementary Fig. 9).

### Robust coding for non-clustering composite letter grouping

In conventional DNA storage systems using only natural bases, sequencing reads exhibit high homogeneity. This enables the use of sequence-level redundancy and clustering with consensus calling to reduce errors[37,38]. However, this strategy fails for composite DNA strands due to their intrinsic heterogeneity (Fig. 2a). To quantify this issue, we measured the number of ordinary strands corresponding to each composite strand. Analysis results revealed an exceptionally high sequence diversity, estimated between $10^{14}$ and $10^{20}$ (Supplementary Table 3). At this scale, the traditional clustering-based error correction scheme becomes computationally infeasible. To enable non-clustering composite letter grouping and correct letter-level errors, we designed reliable double-end indices and applied efficient error correction codes to the payload (Supplementary Note 1).

First, we designed double-end indices to improve the reliability of read identification. The 7-bit address was encoded into two 8-nt natural DNA sequences using Bose-Chaudhuri-Hocquenghem (BCH) and repetition encoding, forming forward and reverse indices (Fig. 2b). Specifically, the forward index was encoded using BCH(15, 7) with an additional parity check bit for error detection. The reverse index employed a parity checksum and a cyclic left-shift mechanism. These encoded double-end indices enable the elimination of crosstalk between synthesis sites.

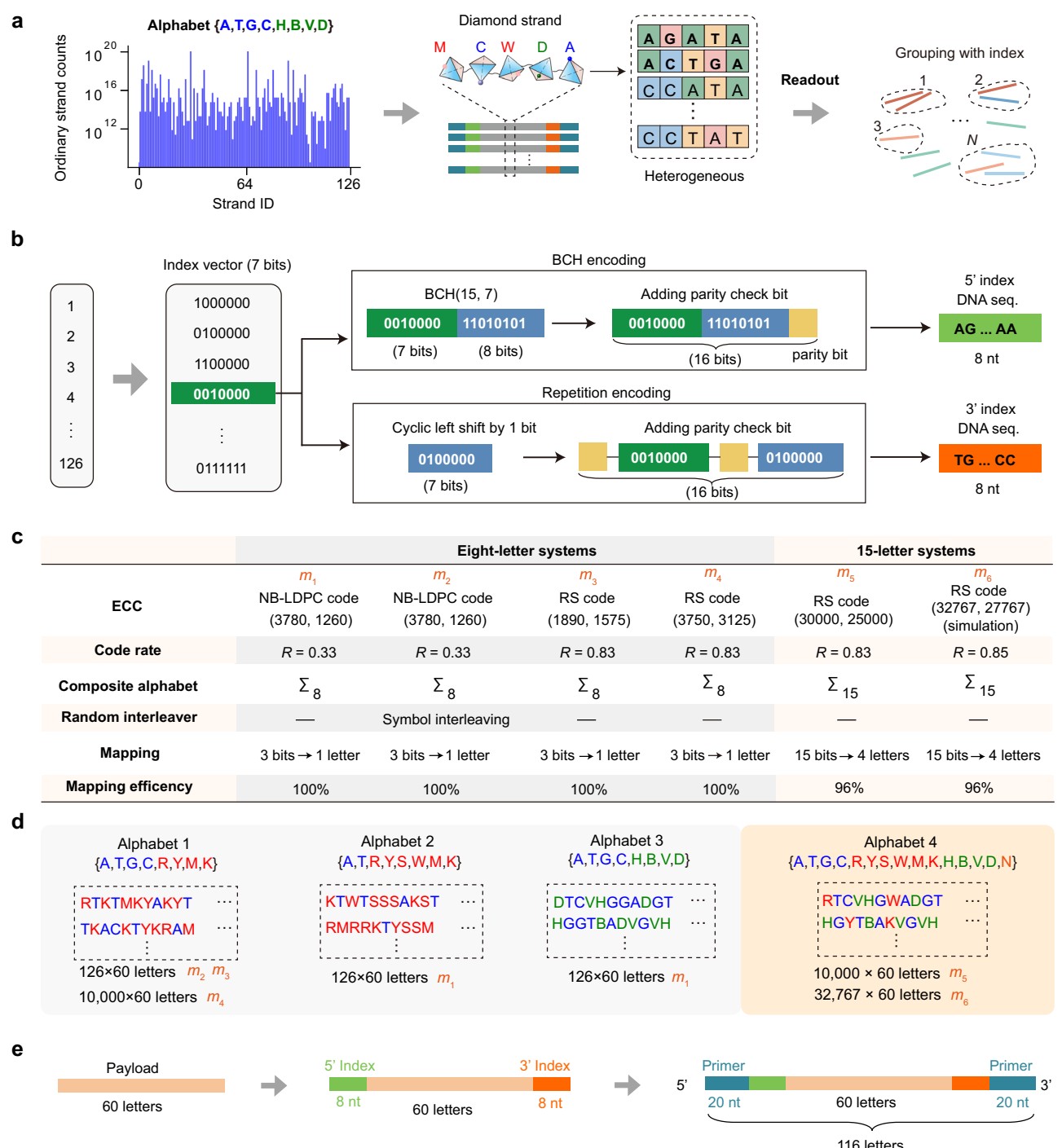

**Fig. 2 | Robust coding methods for composite DNA storage. a** The high heterogeneity of composite strands hinders clustering-based read grouping, motivating index-based read identification. **b** Schematic of the index coding scheme. **c** Overview of tested encoding schemes. NB-LDPC code (code rate = 0.33) and RS code (code rate = 0.83) were used for eight-letter systems, while RS(30000, 25000) was employed for the 15-letter alphabet. **d** Experimentally, different composite alphabets were selected. **e** Composite strand structure in experiments. The primers and indices consist solely of natural bases, while the payload contains composite letters.

Then, we verified the applicability of discrete-entropy composite letters by testing multiple storage schemes across different coding schemes and alphabets (Fig. 2c, Supplementary Figs. 10 and 11, Supplementary Table 4). The encoding and writing process of the composite DNA storage system included three stages: data encoding, composite letter mapping, and DNA synthesis.

For data encoding, in the eight-letter system, 945 bytes of text were first encoded using NB-LDPC(3780, 1260) code ($R = 0.33$) over Galois Field ($GF(64)$), with interleaving to mitigate burst erasures[39]

(Supplementary Fig. 12). To validate high logical density, 2,355 bytes of text data were encoded with RS(1890, 1575) code ($R = 0.83$) over $GF(2^{12})$, achieving 2.5 bits/letter (Supplementary Fig. 13). We further extended the eight-letter scheme to large-scale pools, where a 187,500-byte image file was encoded using RS(3750, 3125) code (Supplementary Fig. 14). In the 15-letter system, both experiments and simulations were performed. A 234,375-byte image file was encoded with RS(30000, 25000) code ($R = 0.83$, 3.125 bits/letter) for experimental verification (Supplementary Fig. 15). A 780,713-byte image file was

encoded with RS(32767, 27767) code ($R = 0.85$, 3.18 bits/letter) for simulations (Supplementary Note 2, Supplementary Fig. 16).

For composite letter mapping, we chose four composite letter alphabets (Fig. 2d) and analyzed the impact of different bit group sizes on mapping efficiency (Supplementary Figs. 17 and 18). The eight-letter alphabet was implemented by converting every three bits into a single composite letter. The mapping efficiency is 100%. For the 15-letter alphabet, a group of 15 bits was converted into four composite letters, corresponding to an optimized efficiency of 96%. We chose the group size of 15 for satisfactory efficiency and error propagation within the group. Details can be found in Supplementary Note 3. Finally, encoded letters were partitioned into strands of 60 letters, padded with indices and primers, generating pools of either 126 or 10,000 strands for synthesis validation (Fig. 2e, Supplementary Tables 5 and 6).

For practical writing into DNA, two synthesis platforms were used to test different composite alphabet implementations (Supplementary Fig. 4). First, 126-strand pools were constructed with eight-letter alphabets using column-based phosphoramidite synthesis. Each strand was synthesized in an individual well, with composite letters implemented by volumetric mixing of A, T, C, and G phosphoramidite monomers. Second, large-scale pools containing 10,000 composite strands were synthesized by array-based inkjet phosphoramidite chemistry. In all experiments, payloads were uniformly 60 letters, flanked with double-end indices and PCR primer sequences (Fig. 2e). For the column-based pools of 126 strands, each carried two indices (8 nt) and two primers (20 nt) at the 5′ and 3′ ends. For the array-based pools of 10,000 strands, the indices were 6 nt or 12 nt at the two ends, respectively.

## A two-stage composite letter detection via discrete information entropy

Accurate data readout in composite DNA storage requires inferring the intended composite letter at each synthesis site based on observed frequencies of constituent natural bases. However, this task is complicated by sampling bias and letter degradation. Inspired by SP in communication modems, we developed a two-stage detection method that exploits the fact that the 15 composite letters exhibit only four distinct information entropy values (Supplementary Notes 4 and 5).

The proposed framework integrates entropy-based letter partitioning with likelihood-based estimation, which reduces the size of the candidate letter set and thereby improves detection accuracy (Fig. 3a, b). In the first stage, we calculated the information entropy of observed base frequencies at each synthesis site and assigned the signals to one of four subsets. Raw sequencing data at various coverages revealed that sampling depth significantly influences the accuracy of observed base frequencies (Supplementary Figs. 19–21). For large-scale pools (10,000 composite strands) with eight-letter or 15-letter composite alphabets, the base frequencies were still guaranteed (Supplementary Figs. 22–25). Nevertheless, the transformation from raw signal distributions to entropy values enabled clear discrimination between composite letter types (Fig. 3c and Supplementary Fig. 26). In the second stage, MLE was applied within each subset to identify the most probable letter. The inferred probabilities were normalized and visualized using radar plots, where the radial distance from the origin quantified divergence from the candidate letters (Supplementary Fig. 27). Optimal letter detection was achieved by comparing with predetermined letters in each subset.

To assess the performance, we first developed a simulation model capable of generating base-frequency distributions, sequencing coverage, and user-defined random errors (Supplementary Fig. 28). The coverage parameter in the model was obtained by fitting experimental copy number distributions at different average sequencing depths. For column-based synthesis pools, the observed read counts per strand followed a negative binomial distribution (Supplementary Figs. 29 and 30). Consistently, in array-based synthesis pools

containing up to 10,000 oligonucleotides, the read count distributions also conformed to the negative binomial model (Supplementary Figs. 31–33). This reflects stochastic sampling and strand-specific differences in PCR amplification efficiency.

Then, we evaluated composite letter detection performance. Simulated sequencing datasets under varying coverage were then processed through our standard readout pipeline. Error rate analysis at both detection stages revealed that most inference errors arose during the SP phase and were more prominent at low coverage (Supplementary Fig. 34). Detection accuracy within each subset improved as coverage increased. To further investigate misclassification behavior, we analyzed spatial distances among the 15 letters in the DNA diamond space and observed that misidentification was strongly correlated with spatial distance (Supplementary Fig. 35).

Finally, we compared the end-to-end recovery performance of our SP method with the KL method[19] and the maximum a posteriori probability (MAP) method[29] with computer simulations (Fig. 3d, e). MAP performs global inference and shows the highest detection accuracy but the longest runtime. SP reduces the candidate space through partitioning, resulting in a slight degradation in accuracy, as well as a reduction in computational complexity. Both SP and MAP methods achieved lower error rates than the KL method. With 1000 independent trials using 16 threads on the same server (Intel Xeon Gold 5220R CPU@2.20 GHz and 256 GB RAM), SP consumed only 25% of the time compared to the MAP method. In the simulation settings, MAP achieved error-free recovery at an average coverage of 25×, SP at 30×, whereas KL required 45× coverage. The performance in vitro experiments will be further verified in the following sections.

## Performance evaluation on constituent base-frequency variation at low coverage

The accuracy of composite letter detection is sensitive to sequencing coverage, as lower coverage leads to increased noise in the observed base frequencies. In particular, insufficient sampling may lead to signal degradation of composite letters. We evaluated the robustness of our detection method at low coverage using synthetic pool datasets, focusing on entropy distribution, error rate, and recovery performance.

First, lower sequencing coverage significantly impairs composite letter detection accuracy due to increased noise in observed base frequencies. Downsampling experiments were performed at coverages ranging from 300× to 10× (Fig. 4a), revealing that lower coverage amplifies sampling noise and broadens the entropy distributions of composite letters. This overlap between distributions decreases their separability (Fig. 4b). To quantify this signal degradation, we evaluated letter detection error rates using three different eight-letter composite alphabets. As expected, lower coverage resulted in increased errors across both stages of the two-stage detection process (Supplementary Fig. 36). Experimental results showed substitution error patterns that closely matched simulation predictions (Supplementary Figs. 37–39). Notably, errors predominantly arose from substitutions between composite letters and their constituent natural bases, while within-subset substitutions were rare. Across all tested alphabets, composite letters were more prone to substitution errors than natural bases (Fig. 4c). Moreover, letters composed of more complex mixtures exhibit higher error rates at the same sequencing coverage (Supplementary Figs. 40 and 41). Although all three eight-letter alphabets had the same size and comparable logical density, Alphabet 1, which contains a higher proportion of natural bases, exhibited significantly lower letter error rates.

Then, the minimum sequencing coverage for data recovery was evaluated with 1000 independent trials on two sequencing datasets. Both the erasure error rate and the letter error rate progressively decreased as the average coverage increased (Fig. 4d, e). At low sequencing coverage, erasures dominated, while with increasing

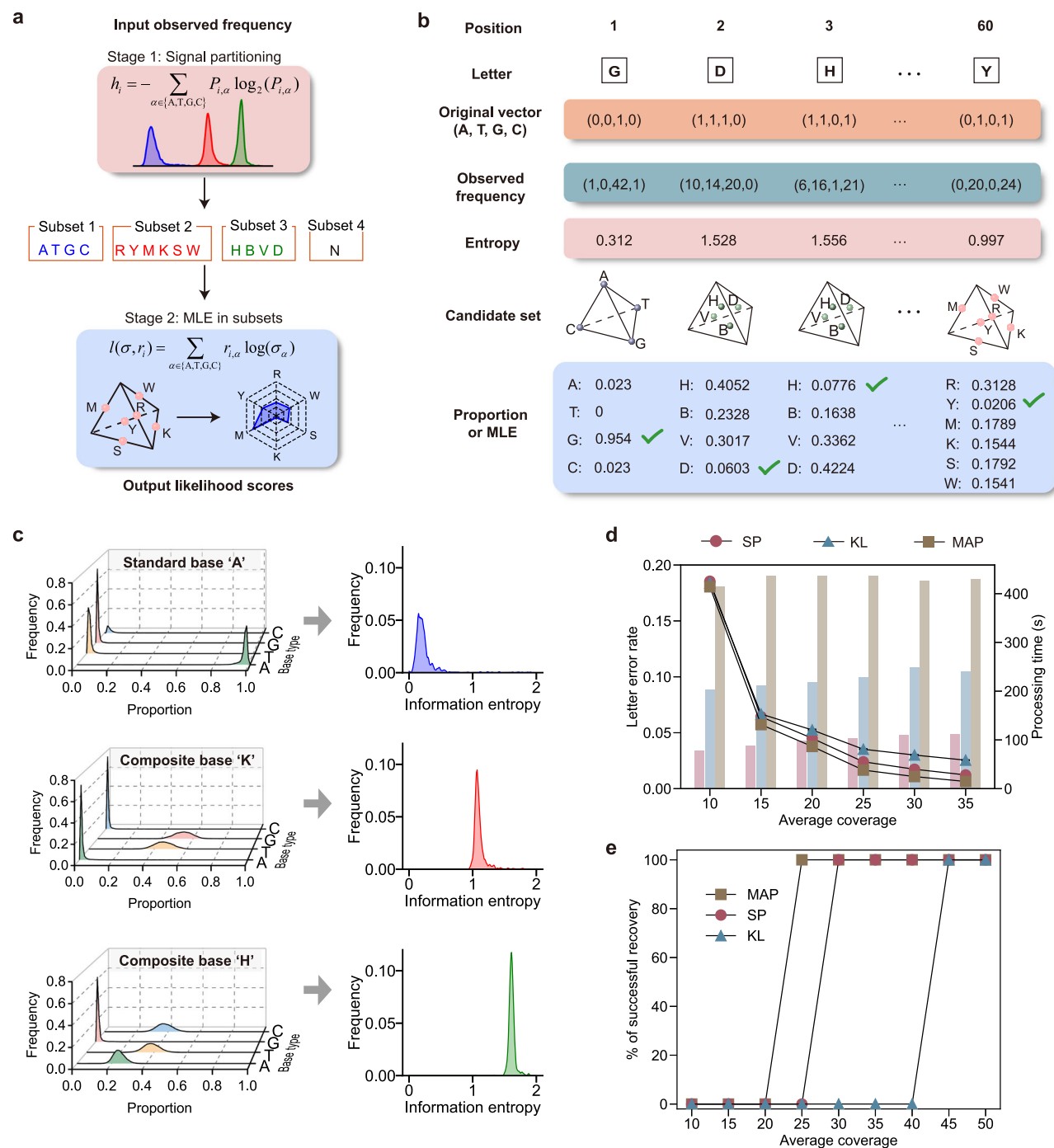

**Fig. 3 | Composite letter detection using set partitioning. a** Workflow of the proposed two-stage composite letter detection method. The first stage involves set partitioning, followed by maximum likelihood estimation in the second stage. **b** Schematic of the composite letter detection process at each position of the sequencing reads. **c** Illustration of information entropy computation based on the observed constituent base frequencies at an average coverage of 500×.

**d** Performance comparison of set partitioning with the benchmark methods, KL divergence, and MAP, for letter detection ($n = 1000$ trials). Line plots show letter error rates (mean ± SD), and bar plots indicate cumulative processing time across 1000 trials. **e** Data recovery performance under varying sequencing coverage ($n = 1000$ trials). Source data are provided as a Source data file.

average coverage, erasures were suppressed, enabling reliable recovery. Using the NB-LDPC code with a code rate of 0.33, error-free recovery was achieved at an average coverage of 9×, even with an erasure error rate as high as 7% (Fig. 4f). Using the RS code with a code rate of 0.83, the coverage of 14× was required for error-free recovery.

Third, we benchmarked the SP method against the KL and MAP approaches at low sequencing coverages. In the column-based eight-letter experiments and the array-based experiments of 10,000 strands (including two eight-letter and two 15-letter systems), MAP consistently achieved the highest detection accuracy (Supplementary Tables 7 and 8). SP, as the second-best in accuracy, showed clear advantages in runtime over both MAP and KL (Supplementary Figs. 42 and 43). For the eight-letter pool of 10,000 strands, MAP enabled error-free recovery at 15× coverage, compared with 16× for SP. Executed with 10 threads, SP completed 300 validation runs in ~10 s (Supplementary Figs. 44 and 45). For the 15-letter system, MAP also

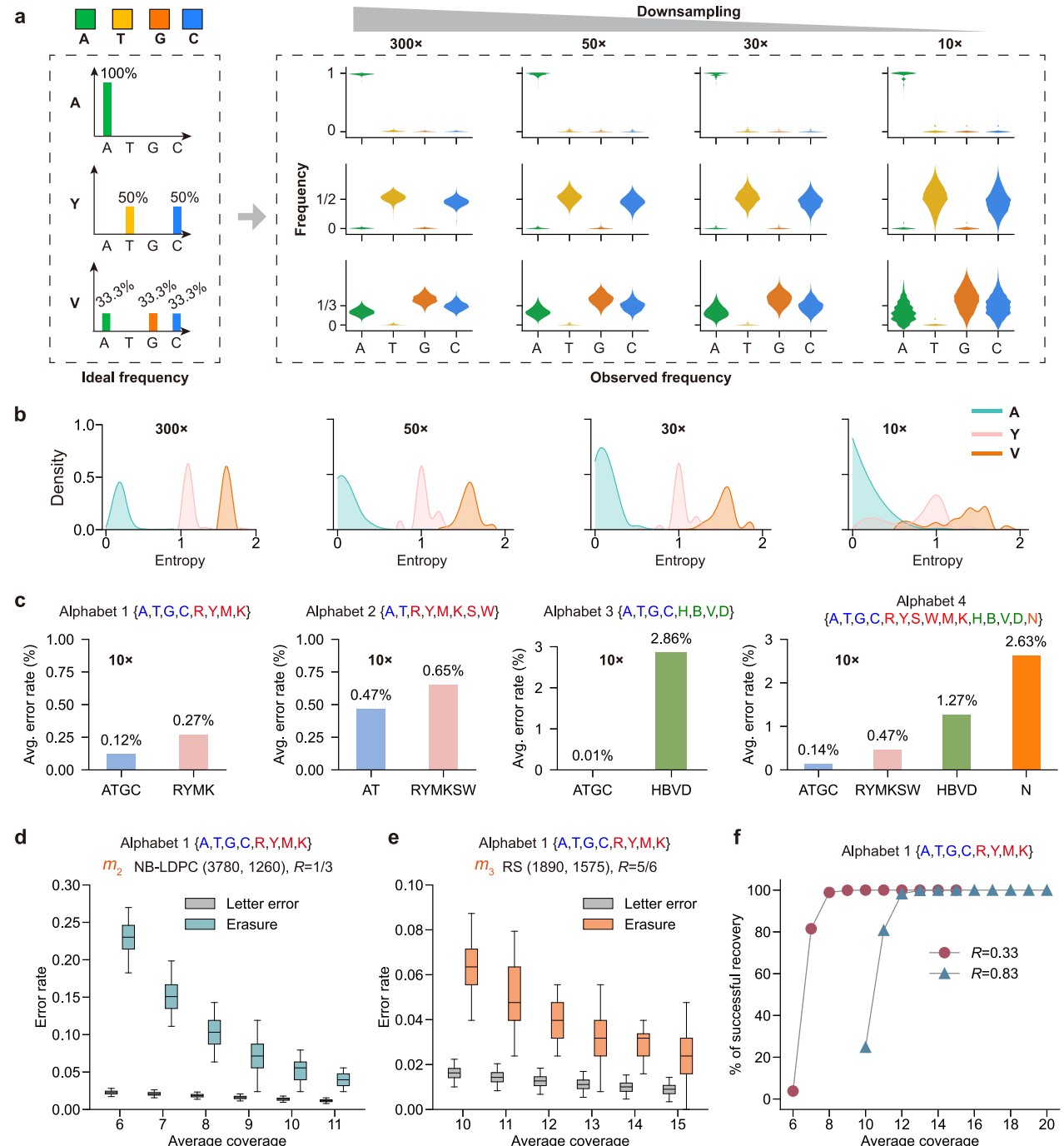

**Fig. 4 | Performance evaluation at low sequencing coverage. a** Coverage affects observed base frequencies; lower coverage increases sampling noise ($n = 954$ for "A," $n = 884$ for letter "Y," $n = 939$ for "V"). Violin widths indicate kernel density estimates; whiskers show the full range. **b** Entropy overlap increases below 10× coverage, impairing subset separability. **c** Letter detection errors at 10× coverage across four alphabets. **d**, **e** Erasure and letter errors decrease with increasing coverage ($n = 1000$ trials). Box plots show the median (center line), interquartile range (box limits, Q1–Q3), and whiskers extending to the most extreme values within 1×IQR. **f** Comparison of data recovery under varying coverage for two encoding schemes. LDPC achieves error-free recovery in all 1000 trials at 9× coverage, whereas the high-rate RS code requires 14× coverage. Source data are provided as a Source data file.

provided the highest accuracy (Supplementary Figs. 46 and 47), while SP showed boundary-related precision loss yet still achieved error-free recovery at 33× coverage, consistent with simulations.

Finally, we applied symbol-level interleaving to improve the error correction capability. Low sequencing coverage not only introduces fluctuations in observed base frequencies but also causes molecule loss due to random sampling bias. The resulting consecutive erasure errors within codewords increase the difficulty of data recovery.

Simulations confirmed that random interleaving disperses burst errors and enhances decoding robustness (Supplementary Figs. 48–51). Recovery tests showed that NB-LDPC codes with interleaving achieved error-free decoding at 9× average coverage, whereas RS codes with a higher code rate (0.83) required 17× average coverage (Supplementary Fig. 52). In contrast, a non-interleaved control based on Alphabet 3 showed poor resilience to molecule loss, requiring much higher coverage for reliable recovery (Supplementary Fig. 53). Additionally,

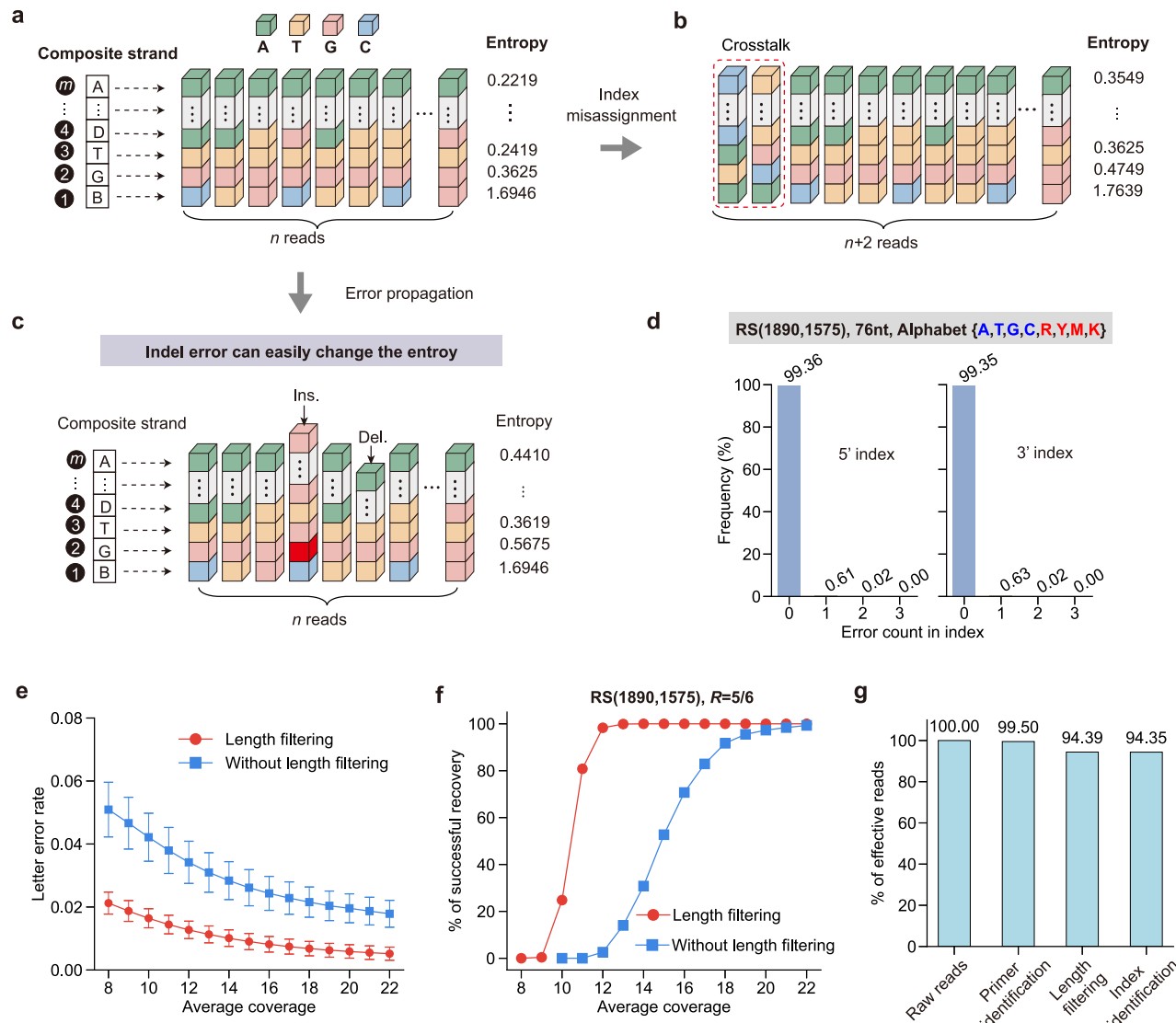

**Fig. 5 | Performance evaluation considering interference and indels. a** Example of information entropy calculation based on constituent base frequencies. **b** Illustration of the information entropy variation due to interference reads (crosstalk). **c** An example shows the impact of indel error propagation on information entropy. **d** The percentage of errors in indices. More than 99% of reads had error-free indices. **e** Length filtering prevents error propagation and improves letter recovery in the experimental data from Alphabet 2 was further impaired by the synthesis failure of a single composite strand.

detection accuracy ($n = 1000$ trials; mean ± SD). **f** Length filtering reduces the sequencing coverage required for data recovery. Plots show the number of successful recoveries out of 1000 independent trials at each coverage. **g** The proportion of effective reads after primer identification, length filtering, and index identification. Source data are provided as a Source data file.

**Performance evaluation on strand crosstalk, error propagation, and DNA degradation**

The reliability of composite letter detection is crucial for accurate data recovery in DNA storage. However, three factors compromise performance: crosstalk caused by interference reads, error propagation introduced by indels, and strand degradation. Crosstalk stems from the potential assignment errors between the indices, leading to misidentification of strands and altered entropy values (Fig. 5a, b). Indel-induced propagation errors distort the length and structure of reads, further interfering with the letter detection process (Fig. 5c). Notably, homopolymers are unavoidable in sequencing reads, and long homopolymers may introduce more indels (Supplementary Fig. 54). Fortunately, long homopolymers occurred rarely. Homopolymer length distributions in composite systems are similar to four-letter DNA storage without homopolymer optimization or random sequences (Supplementary Fig. 55). Several DNA storage systems guaranteed homopolymer length through appropriate encoding, e.g., rotation coding[4] or homopolymer filtering[9]. We did not apply homopolymer optimization, balancing the efficiency and performance[40]. To address these challenges, we incorporated two tailored strategies into the readout pipeline: robust double-end indices to mitigate strand crosstalk and DNA degradation; length filtering to reduce the impact of indel errors during synthesis and sequencing.

First, based on paired-end sequencing data, we established a pipeline that stitches read pairs, aligns primers, and extracts the central region containing the index and payload (Supplementary Figs. 56–58). The retained payload regions, which may contain errors, were grouped based on matching index sequences, each representing multiple copies of a unique payload. Analysis revealed that 99% of indices in our sequencing reads were error-free, and the remaining errors involved fewer than two substitutions (Fig. 5d and Supplementary Fig. 59). We retained reads in which both ends contained

fewer than two substitution errors and were mapped to the same reference composite strand.

Then, the double-end strand structure enables robust data retrieval even when partial degradation occurs, for example, a strand break. Any fragment retaining at least one intact primer–index pair can still be correctly clustered and aligned according to its index (Supplementary Fig. 60). To evaluate the performance on strand break, we performed accelerated thermal aging experiments. The composite pool containing 126 strands of 116 letters was incubated at 80 °C (2.5 h), 85 °C (2 h), 90 °C (2 h), and 95 °C (2 h), with an unheated sample as the control. Under higher temperatures, the read lengths showed a pronounced shift toward shorter fragments. Consistently, the proportion of correctly assembled 116 nt reads decreased, while truncated fragments accumulated (Supplementary Figs. 61–63). At 95 °C, nearly 14% of reads contained only a single-end primer (Supplementary Fig. 64). Despite this degradation, data recovery remained highly robust. Letter error rates increased moderately with degradation temperature. Error-free recovery was still achieved at sufficient sequencing coverage (Supplementary Fig. 65). Stress tests using short fragments revealed that the proposed scheme could recover the original data error-free, though the required sequencing coverage was increased (Supplementary Fig. 66).

Finally, length filtering was introduced to suppress error propagation caused by indels. Under the same average coverage, we evaluated the effectiveness of length filtering in reducing the symbol error rate by comparing two schemes. Across both coding schemes ($R = 0.83$ and $R = 0.33$), filtering consistently improved letter detection accuracy (Fig. 5e, f and Supplementary Fig. 67). At the coverage of 14×, filtering reduced the average composite-letter error rate from 2.8% to 1.0% (accuracy >99%). With the RS code ($R = 0.83$), length filtering enabled error-free recovery in all trials. Without filtering, only ~30% of trials achieved error-free recovery under the same coverage. Although filtering decreased the number of usable reads, 94.39% were retained after primer identification, length filtering, and index identification, ensuring high-quality input for downstream detection (Fig. 5g). Supplementary Table 9 presents the results of all pools, including small-scale pools and large-scale pools.

## Cost analysis on composite letter DNA storage at low coverage

We performed a cost analysis in two application modes, consistent with prior studies, quantifying both synthesis and sequencing costs for our eight-letter and 15-letter schemes (Supplementary Note 6). Synthesis cost was modeled from logical density and sequencing cost from average coverage[19,25].

First, we estimated the total cost of storing 1-MB user data in DNA, including synthesis and sequencing. To compute the overall storage cost of 1-MB user data, we applied unit prices of $0.05/100 nt for synthesis[9] and $0.0000012/100 nt for sequencing[41] (Supplementary Fig. 68). We estimated the nucleotides needed to store 1-MB user data. Under these assumptions, the synthesis-to-sequencing cost ratio can be as high as 40,000:1, with the synthesis remaining the dominant cost. On the one hand, only considering the payload part, our eight-letter and 15-letter systems reduced the synthesis cost by 47% and 57%, respectively, compared with the conventional four-base DNA storage system[25]. On the other hand, considering the index overhead (excluding primers), for a similar alphabet size of six or eight, we obtained similar cost reductions. For the 15-letter alphabet, we achieved large-scale validation using 10,000 strands, also with a slight increase in synthesis cost (Supplementary Fig. 68). This was primarily due to the adoption of a more robust double-end index scheme that introduced additional redundancy to improve reliability.

Then, as synthesis technology continues to improve and costs decline, we further normalized total cost under representative synthesis-to-sequencing ratios (e.g., 1000:1 and 500:1) relative to a baseline natural-base system with 5× coverage[25] (Supplementary Table 10). Only considering the region excluding primers at a 1000:1 ratio, the eight-letter and 15-letter systems reduced the total storage cost by 47% and 56%, respectively. When the ratio was reduced to 500:1, the total cost reductions remained at 46% and 55%, respectively.

## Discussion

Compared with existing composite DNA storage methods, our approach introduces several key innovations. First, we adopt a signal design framework rooted in classical communication theory to construct a decomposable constellation of composite letters. Second, we develop a robust two-stage signal detection scheme based on entropy-guided SP, which enhances letter detection accuracy within each subset by grouping frequency patterns into different subsets. This design fundamentally differs from previous composite systems that relied on observed frequencies or static probability thresholds. Third, recognizing the limitations of clustering methods for composite strands, we encode double-end indices to improve the reliability of read identification and introduce a length filtering step during readout. These two tailored strategies reduce crosstalk caused by interfering reads and mitigate error propagation induced by indels. We experimentally demonstrated the feasibility of composite DNA storage under low sequencing coverage. In a small-scale setting, 126 composite strands were synthesized using column-based chemistry for proof-of-concept verification. In a large-scale setting, four pools, each containing 10,000 strands, were synthesized using array-based inkjet chemistry. Using an eight-letter alphabet, we achieved 2.5 bits per letter and error-free data recovery at an average sequencing coverage of 14×. Notably, the 15-letter alphabet was experimentally realized at the 10,000-strand scale, achieving 3.125 bits per letter for payload (or 2.23 bits per letter for payload plus indices) and error-free recovery at an average sequencing coverage of 33×. This low-coverage experimental validation of a high-cardinality composite system surpassed earlier six-letter or 15-letter schemes[18,19].

Increasing logical density is essential for reducing DNA storage costs. In our experiments, we showed that for a fixed alphabet size, the logical density remains constant, but the letter error rate is strongly influenced by the composition of the composite letters. Among the three eight-letter alphabets with identical density, those containing more natural bases or fewer constituent bases per letter showed significantly lower error rates. This finding provides practical guidance for the future design of composite alphabets in DNA storage systems. Another approach to increasing logical density involves expanding DNA alphabets through the incorporation of chemically modified nucleotides[15] or non-natural nucleotides[16,17]. In the future, integrating composite letters with expanded base sets could further enhance storage density. Our proposed signal model is readily adaptable to such extensions. If the base number increases from 4 to 8, the logical density with composite letters is expected to achieve an improvement from $\log_2(2^4 - 1) \approx 3.9$ to $\log_2(2^8 - 1) \approx 8.0$ per synthesis cycle, even if only the decomposable constellation is used. Moreover, if our decomposable constellations are employed, practical and reliable coding schemes are expected to improve logical density.

While our entropy-guided SP performs well for small alphabets (up to 15 letters), its resolution declines as constellation size increases, due to overlapping entropy distributions and rising misclassification rates. On the one hand, one promising direction is to move from entropy-based decoding to information-theoretic coding over probability spaces. For example, limited-magnitude probability error (LMPE) models, which treat letter errors as bounded distortions in high-dimensional probability vectors[31], provide a promising theoretical foundation. Codes optimized under LMPE constraints may enable robust detection even with larger composite alphabets. On the other hand, all these approaches remain vulnerable to indel errors. In particular, low-cost synthesis techniques are prone to indel errors, and rapid nanopore sequencing is also inherently indel-prone[42,43]. These

errors can alter the observed frequencies of constituent bases, thereby compromising the accuracy of the composite letter detection. In this work, we employed length filtering to alleviate this adverse effect. Future research should focus on more practical error models and more efficient error correction schemes[44,45], with the ultimate goal of improving the reliability of general composite DNA storage systems.

## Methods

### A decomposable diamond for composite letter DNA storage

A decomposable diamond model was proposed to represent the composition of composite letters and partition the 15 decomposable constellations into four different subsets.

- Subset 1 (Vertices): this subset consists of the four natural bases, {A, T, G, C}, positioned at the vertices of the diamond constellation model.
- Subset 2 (Edges): defined by the six central points on the edges of the diamond, this subset includes {R, M, Y, K, S, W}. Each composite letter in this group is synthesized as an equimolar mixture of any two natural bases.
- Subset 3 (Faces): corresponding to the four central points on the faces of the diamond, this subset comprises {H, B, V, D}, where each composite letter is formed as an equimolar mixture of any three natural bases.
- Subset 4 (Centroid): representing the center point of the diamond model, this subset consists of a single composite letter {N}, synthesized as an equimolar mixture of all four natural bases.

Each letter in the diamond model is characterized by a probability vector $\sigma(\sigma_A, \sigma_T, \sigma_G, \sigma_C)$ that represents the projection relationship between natural bases and composite letters (Supplementary Table 11). Zero-probability events were replaced with a smoothing term ($\epsilon = 10^{-6}$) to ensure numerical stability, where $\sigma_A + \sigma_T + \sigma_G + \sigma_C = 1$.

### Encoding scheme for experimental proofs with eight-letter and 15-letter alphabets

Two sets of experiments on the composite-letter DNA storage were conducted based on the constructed composite-letter alphabets (Supplementary Data 1). We devised four alphabets as follows.

- Alphabet 1: {A, T, G, C, R, Y, M, K};
- Alphabet 2: {A, T, R, Y, M, K, S, W};
- Alphabet 3: {A, T, G, C, H, B, V, D};
- Alphabet 4: {A, T, G, C, R, M, Y, K, S, W, H, B, V, D, N}.

In the small-scale composite strand storage, three eight-letter alphabets (Alphabets 1–3) were employed. Correspondingly, three different coding schemes were used. They were the NB-LDPC(3780, 1260) over $GF(64)$, the NB-LDPC(3780, 1260) over $GF(64)$ with random symbol interleaving, and the RS(1890, 1575) over $GF(2^{12})$. Each encoding process produced 126 composite strands of 60 letters, which were synthesized into four oligonucleotide pools. In the eight-letter system, every 3 bits were converted into one composite letter, and when encoded with RS(1890, 1575) at a code rate of 0.83, a logical density of 2.5 bits per letter was achieved (excluding indices and primers).

In the large-scale composite strand storage, an eight-letter alphabet (Alphabet 1) and a 15-letter system (Alphabet 4) were tested. For the eight-letter system, a 187,500-byte digital file was segmented into 40 data blocks and encoded using RS(3750, 3125) over $GF(2^{12})$, yielding 10,000 composite strands. For the 15-letter system, a 234,375-byte image file was segmented into 5 data blocks and encoded using RS(30000, 25000) over $GF(2^{15})$, yielding 10,000 composite strands. Every 15 bits was mapped to four composite letters, corresponding to a logical density of 3.125 bits/letter (excluding indices and primers). In both systems, each payload strand consisted of 60 composite letters flanked by indices and primers.

### Double-end indices for homology identification of sequencing reads

To minimize the interference of crosstalk during data readout, double-end indices were designed to label the order of composite strands used in experimental validation. These indices consisted exclusively of natural bases and were generated through the following steps:

1) A 7-bit address in the range [1, 126] was assigned to each of the 126 DNA sequences.
2) The 5′ end index was generated using a BCH(15, 7) code with an additional parity check bit. The encoded sequence was mapped to DNA bases using the rule: 00→A, 01→T, 10→G, 11→C.
3) For the 3′ end index, a parity check bit was first generated based on the original 7-bit address. This address was then circularly left-shifted by one bit to generate a second 7-bit sequence. The 16-bit index was structured as follows: parity bit–original bits–parity bit–left-shifted address bits.
4) Following the same mapping rule as in Step 2, the encoded address bit sequence obtained in Step 3 was converted into an 8-nt index DNA sequence and appended to the 3′ end of the composite strand.
5) Each payload was flanked with an 8-nt index and a 20-nt amplification primer at both ends, resulting in a total sequence length of 116 letters.

### Synthesis of small-scale pools (column-based synthesis)

A total of four small-scale composite-letter oligonucleotide pools were prepared. For each pool, 126 composite DNA sequences were synthesized using an Oligo-192 DNA Synthesizer (Hippobio, Huzhou, China). Each strand was synthesized independently in a dedicated column, with composite letters implemented through volumetric mixing of A, T, G, and C phosphoramidite monomers. Oligonucleotides were purified by high-performance liquid chromatography (HPLC). All 126 samples were then manually combined in equimolar amounts to generate one master pool. Four such master pools were obtained, with final DNA concentrations of 2482, 2515, 301, and 459 ng/μL, respectively.

### Synthesis of large-scale pools (array-based inkjet printing synthesis)

Four large-scale oligonucleotide pools were designed, each comprising 10,000 composite strands: two based on the eight-letter alphabet and two based on the 15-letter alphabet. These pools were synthesized by Dynegene Technologies (Shanghai, China) using an array-based inkjet printing synthesizer. During synthesis, four inkjet nozzles dispensed monomer droplets onto array chips according to predefined base-mixing ratios. The eight-letter system used the alphabet {A, T, G, C, R, Y, M, K}, and the 15-letter system used the alphabet {A, T, G, C, R, M, Y, K, S, W, H, B, V, D, N}. Each large-scale pool was delivered as a lyophilized master pool containing 3 μg of DNA.

### Library preparation of small-scale pools

All four small-scale pools were diluted to 0.1 ng and amplified in 25 μL PCR reactions using Phusion High-Fidelity DNA Polymerase (Hot Start Flex 2X Master Mix, New England Biolabs, Cat# M0536S) (Supplementary Table 12). The PCR program included an initial denaturation at 98 °C for 45 s, followed by 30 cycles of 98 °C for 10 s, 60 °C for 30 s, and 72 °C for 30 s, and a final extension at 72 °C for 1 min. Amplification products were purified using AMPure XP Beads (Beckman Coulter, Cat# A63880) at a 3:1 bead-to-DNA ratio and eluted in 20 μL of double-distilled water (ddH2O).

Furthermore, two small-scale pools designed with Alphabet 1 {A, T, G, C, R, Y, M, K} were subjected to low-cycle PCR with the KAPA HiFi HotStart PCR Kit (Roche, Cat# KK2502). The PCR program consisted of an initial denaturation at 98 °C for 45 s, followed by 10 cycles of 98 °C for 10 s, 60 °C for 30 s, and 72 °C for 30 s, with a final extension at 72 °C

for 1 min (Supplementary Table 13). Each 25 μL reaction contained 0.8 ng of DNA sample, 0.75 μL of KAPA dNTP Mix (10 mM each), 1.5 μL of 10 μM forward primer, 1.5 μL of 10 μM reverse primer, 5 μL of 5× KAPA HiFi Fidelity Buffer, and 0.75 μL of KAPA HiFi HotStart DNA Polymerase (1 U/μL). The final volume was adjusted to 25 μL using ddH2O. PCR products were purified using AMPure XP Beads (Beckman Coulter, Cat# A63880) at a 1.8× volume ratio and quantified using the Qubit 1X dsDNA HS Assay Kit (Thermo Fisher Scientific, Cat# Q33231).

### Library preparation of large-scale pools
Libraries from four large-scale pools were PCR amplified for 10 cycles using the KAPA HiFi HotStart PCR Kit (Roche, Cat# KK2502) following the same reaction composition and thermal cycling parameters (Supplementary Table 13). PCR products were purified using AMPure XP Beads (Beckman Coulter, Cat# A63880) at a 1.8× volume ratio. Purified DNA was eluted in ddH2O and quantified using the Qubit 1X dsDNA HS Assay Kit (Thermo Fisher Scientific, Cat# Q33231).

### Sequencing and data generation
After PCR amplification, purified PCR products were sequenced by Novogene (Beijing, China) and LC-Bio Technology Co., Ltd. (Hangzhou, China). The two small-scale pools (based on Alphabet 1) and the four large-scale pools (10,000 strands each) were sequenced by Novogene on the Illumina NovaSeq X Plus platform, generating 150-nt paired-end (PE150) raw reads. In parallel, for cross-platform validation, all four small-scale pools were also sequenced by LC-Bio Technology using Illumina sequencing platforms.

### Accelerated aging test of oligo pools
Accelerated aging experiments were performed on a small-scale pool consisting of 126 composite strands encoded with the RS code. Four identical samples, each 25 μL, were incubated at 95 °C for 2 h, 90 °C for 2 h, 85 °C for 2 h, and 80 °C for 2.5 h in the absence of light and then stored at −20 °C until sequencing.

DNA concentrations before and after aging were measured using the Qubit 1X dsDNA Assay Kit (Thermo Fisher Scientific, Cat# Q33231). Quantification revealed initial DNA masses of 57, 66, 59, and 82 ng, which decreased to 34, 35, 41, and 49 ng, respectively, after accelerated aging. The final four degraded DNA libraries and the non-degraded control DNA libraries were sequenced by Novogene on the Illumina NovaSeq X Plus, generating 150-nt paired-end (PE150) raw reads.

### Simulation verification of the information density
To evaluate the information density of composite letter DNA storage, we conducted simulations using the RS(32767, 27767) code over $GF(2^{15})$ with a code rate of 0.85 for a 15-letter alphabet. Encoded RS codewords were segmented into 15-bit blocks, each mapped to four base−15 symbols, achieving a mapping efficiency of 96%. Using this approach, a 780,713-byte image was encoded into 15 RS codewords, generating 32,767 composite strands (60 letters per strand), resulting in a logical density of 3.18 bits per letter.

To assess system performance, a simulation model was constructed. This model involves three parameters: (i) sequencing coverage $D$ for each strand under a fixed average sequencing coverage $M$, (ii) the error rates (insertions, deletions, and substitutions), and (iii) random instantiation as a permutation over A, T, G, and C. The specific simulation process follows these steps:

(1) Based on real sequencing reads, a Poisson-Gamma distribution (equivalent to a negative binomial distribution) was fitted to model the copy number of each sequence under varying sequencing coverage $M$. This modeling resulted in a set of shape parameters $r$ and success probabilities $p$ (Supplementary Table 14). Simulations were performed to determine the number of strand copies under varying coverage conditions, using 32,767 composite strands with a 15-letter alphabet.

(2) Each position containing a composite letter was instantiated as random combinations of A, T, G, and C according to a predefined probability vector $\sigma(\sigma_A, \sigma_T, \sigma_G, \sigma_C)$.

(3) The error model assumed that each position of instantiated DNA sequences was subject to a total error rate $P_{error}$, defined as the sum of the substitution error rate $P_{sub}$, the insertion error rate $P_{ins}$, and the deletion error rate $P_{del}$. For each nucleotide $i$ (where $i \in \{A, T, G, C\}$), the error probability $P_{error\_i}$ was defined as: $P_{error\_i} = P_{ins} + P_{del} + P_{sub}$. Given the total number of positions $k$ and the count of each nucleotide $\alpha_i$, the total expected number of errors across all positions was calculated as:

$$\sum_{i \in \{A, T, G, C\}} \alpha_i \times P_{error\_i} = k \times (P_{ins} + P_{del} + P_{sub}). \quad (1)$$

In practice, we selected three types of errors ($P_{ins} = 0.001$, $P_{del} = 0.001$, $P_{sub} = 0.002$) and performed 1000 repetitions at each sequencing coverage. The simulated data were processed using the same readout pipeline as real sequencing reads.

### A fast data recovery pipeline from paired-end sequencing reads
The data recovery of the composite DNA storage consists of the following steps:

(1) Read assembly: paired-end reads were assembled using the paired-end read merger program PEAR[46], and the target region (including index and payload) was extracted based on the alignment with paired-end primers.

(2) Length filtering: reads that did not match the ideal length for the target region after primer identification were discarded.

(3) Index identification: payload data were identified and grouped using double-end indices. All copies of reads containing the same index sequence correspond to a unique composite strand.

(4) Composite letter detection: a two-stage detection method was applied, incorporating SP and MLE.

(5) Error correction and decoding: the inferred composite letters were decoded and corrected using the LDPC decoding or RS decoding, achieving error-free data recovery.

### Two-stage letter detection method
To accurately recover data, we proposed a two-stage composite letter detection method (Supplementary Notes 4 and 5). In the first stage, the observed base frequencies were mapped to information entropy and partitioned into several subsets. At each synthesis position $i$, the observation probability for each natural base $P_{i,\alpha}$ was defined as

$$P_{i,\alpha} = \frac{r_{i,\alpha}}{\sum_{\alpha \in \{A, T, G, C\}} r_{i,\alpha}} \quad (2)$$

where $r_{i,A}, r_{i,T}, r_{i,G}, r_{i,C}$ are the counts of natural bases at position $i$, and $\sum_{\alpha \in \{A, T, G, C\}} P_{i,\alpha} = 1$. Information entropy for each synthesis position $i$ was defined as

$$h_i = -\sum_{\alpha \in \{A, T, G, C\}} P_{i,\alpha} \log_2(P_{i,\alpha}) \quad (3)$$

where $h_i$ quantifies the uncertainty associated with the observed base frequencies. Each of the 15 composite letters was associated with a theoretical Shannon entropy of 0, 1, 1.5, or 2 bits. The measured entropy of each letter was compared to these four discrete values, and the letter was assigned to the subset corresponding to the closest entropy level:

- $S_1 = \{A, T, G, C\}$, natural bases.
- $S_2 = \{Y, M, K, R, S, W\}$, consisting of two types of natural bases.
- $S_3 = \{H, B, V, D\}$, consisting of three types of natural bases.
- $S_4 = \{N\}$, consisting of four types of natural bases.

In the second stage, a more precise comparison of each candidate letter was performed using a MLE method within each subset, as detailed in the following:

First, the divergence between the observed frequency and the candidate letter was quantified. A likelihood function $l(\sigma, r_i)$ was formulated based on the observed natural base counts at each synthesis position $r_i$ and the original probability vector $\sigma(\sigma_A, \sigma_T, \sigma_G, \sigma_C)$ of each candidate composite letter within the subset. The likelihood function was defined as

$$l(\sigma, r_i) = \sum_{\alpha \in \{A, T, G, C\}} r_{i,\alpha} \log(\sigma_\alpha) \tag{4}$$

Then, the probabilities obtained from the MLE were normalized to determine the composite letter with the highest likelihood as the detection result for each synthesis position.

Finally, the detected composite letter was mapped to its corresponding original bit sequence, followed by an error correction scheme to resolve residual substitution or erasure errors. This process enables accurate reconstruction of the original digital files.

### Statistics and reproducibility

No statistical method was used to predetermine sample size. No data were excluded from the analyses. The experiments were not randomized. The investigators were not blinded to allocation during experiments and outcome assessment.

### Reporting summary

Further information on research design is available in the Nature Portfolio Reporting Summary linked to this article.

## Data availability

The encoded composite letter sequences are available via Zenodo at https://doi.org/10.5281/zenodo.17350307[47]. The sequencing data (FASTQ format) from the array-based synthesis pools have been deposited in the Sequence Read Archive under accession number PRJNA1345374, and are also available via Zenodo at https://doi.org/10.5281/zenodo.17350307[47]. The sequencing data from the column-based synthesis experiments have been deposited in the Sequence Read Archive under accession number PRJNA1258704, and are also available via Zenodo at https://doi.org/10.5281/zenodo.15337157[48]. Source data are provided with this paper.

## Code availability

The source code for composite letter detection and data readout is publicly available and has been deposited in GitHub at https://github.com/TJU-QiGe/Two-stage-composite-letter-detection-method-using-set-partitioning, under MIT license. The specific version of the code associated with this publication is archived in Zenodo and is accessible via https://doi.org/10.5281/zenodo.17905993[49]. This implementation makes use of several third-party software packages under their respective licenses, including RS codes by Morelos-Zaragoza, R. (https://www.eccpage.com), seqtk by Li, H. (https://github.com/lh3/seqtk), and edlib by Šošić, M. (https://github.com/Martinsos/edlib).

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

## Acknowledgements

This work was supported by grants from the National Key R&D Program of China (2023YFA0913800 and 2021YFF1200200 to W.C.; 2024YFF1500500 to Y.Y.). The authors thank Dashun Huang (Hippobio Co., Ltd., Huzhou, China) for assistance with the column-based synthesis of composite DNA strands, and Dynegene Technologies (Shanghai, China) for their support in the array-based synthesis of large-scale composite pools. The authors also thank Lulu Li (LC-Bio Technology Co., Ltd., Hangzhou, China) for support in library preparation and sequencing, and Rui Qin for performing the amplification and accelerated aging experiments.

## Author contributions

W.C. and Y.Y. conceived the project and reviewed the results. Q.G., C.H., and W.C. designed the composite DNA storage system and wrote the encoding and decoding programs. Q.G. developed the two-stage composite letter detection program and performed the simulations and experimental validation. Q.G., M.R., and T.Q. analyzed sequencing data. Q.G. and W.C. validated the results and wrote the manuscript. All authors supervised the results, revised the manuscript, and approved the final manuscript.

## Competing interests

Q.G., C.H., and W.C. have been granted a Chinese patent related to the encoding and decoding approach for composite letter DNA data storage (patent number CN119649874B). The remaining authors declare no competing interests.
