## [Transparent Peer Review file · Nature Communications]

DNA Diamond Formulates a Decomposable Composite Letter Constellation Model for DNA Data Storage

Corresponding Author: Professor Weigang Chen

Version 0:

Reviewer comments:

Reviewer #1

(Remarks to the Author)

This work discussed an interesting approach, named DNA diamond, which is a composite letter constellation model for encoding in DNA data storage. This model expands the letters available for DNA data storage from 4 (simply natural bases A, T, C, G) to 15, considering the probability distribution of natural bases. Compared with other composite letter approaches, this method achieved the highest bit-to-letter coding efficiency with a relatively low sequencing coverage required to fully recover the data. The authors also developed methods to enable accurate and lossless recovery of data, including set partitioning, maximum likelihood estimation (MLE), appending double-end indices and perform length filtering.

The concept of DNA diamond is great. However, in practice, one of the key challenges of DNA data storage is the synthesis cost, which is measured by how many bits are encoded in one nucleotide. Although this work did a great job in improving the letter efficiency, I am not able to find any elaborations on the actual coding potential of this coding strategy. How many nucleotides, in total, are synthesized to encode each experimental (7,560-bit and 18,900-bit data) and simulation (780,713-byte image)? In my understanding, as shown in Supplementary Figure 2, N strands of 60-nt DNA only yielded 60 letters. If 1 letter can encode 3 bits in their wet experiments, then these strands just encoded 180 bits of data. I suggest the authors to calculate and clearly state the coding potential (not considering the indices, PCR primers and error correction codes) and the net information density (all synthesized nucleotides considered) using DNA diamond, with a unit of "bits/nt".

Except the major concern above, I also have other concerns in terms of the feasibility of this coding strategy in real settings:

1. In practice, the existence of homopolymers (3 or more) of natural DNA bases is detrimental to sequencing-based data recovery. However, by mapping 1 letter to 3-bit of data, it is impossible to avoid the appearance of homopolymers in synthesized DNA strands. How would the authors overcome this issue?
2. In their simulation experiment, the mapping between 15 binary bits ($2^{15} = 32,768$) and 4 letters ($15^4 = 50625$) left so many combinations of letters not used in encoding and therefore is not efficient. This originates from the fact that this encoding strategy is very sensitive to the total bits of information to be coded, and the coding efficiency is not always high for different data sizes. In addition, I noticed that for different sizes of data, a different size of block should be carefully selected (6-bit in wet experiment and 15-bit in simulation). Did the authors investigate the impact of the size of block on coding potential?
3. I am not clear on how the authors can accurately control the read count of each DNA strand through sequencing. I assume this was achieved through adjustment of copy numbers of each strand. However, this significantly complicates experimental procedure and is apparently not practical in commercial DNA data storage settings. In addition, the efficiency of PCR is different for different amplicon sequences (the payload in this study). For example, the amplification may be difficult to proceed when there are homopolymers in the amplicon. When this encoding strategy scales up to 10,000 or more strands in one oligo pool (which is very typical in real DNA data storage applications), can the probability distribution of each natural DNA base still be guaranteed?
4. The degradation of DNA is also a concern in practice, and the degradation rate of DNA strands with different sequences may also be non-homogeneous, which causes distortion of the assumed probability distribution of natural bases. I suggest the authors to perform accelerated aging experiments on their synthesized DNA pools (elevated temperatures, acidic and basic conditions, repeated freeze-thaw cycles) to investigate if this encoding strategy is less resistant to DNA degradations compared with conventional encoding strategies.

(Remarks on code availability)

Reviewer #2

(Remarks to the Author)

This manuscript introduces a “diamond” composite-letter alphabet for DNA storage and an entropy-guided, two-stage detection pipeline: composite letters are first partitioned into entropy-defined subsets and subsequently identified within subsets by maximum-likelihood inference. Wet-lab experiments on an eight-letter alphabet report achieving ~2.5 bits per letter and error-free recovery at 18× Illumina coverage. Although these proof-of-concept results are promising, the study currently exhibits several empirical, methodological, and contextual shortcomings.

Major:

1. The manuscript contains several instances of imprecise descriptions and inaccurate data. In particular, the authors should more accurately clarify the challenges of composite-letter storage and carefully contextualize prior studies.

(1) Lines 42-47: While the authors identify key manifestations of the composite-DNA readout challenge—namely, difficulty in discriminating similar base mixtures and obscuration of constituent components by sequencing noise—the discussion remains somewhat surface-level and does not sufficiently articulate the fundamental underlying causes. In fact, the elevated error rates in composite DNA storage arise from two interacting sources: (i) synthesis and sequencing errors, which perturb the intended base mixture proportions; and (ii) statistical sampling bias, especially under low coverage, which leads to fluctuations in observed base-frequency distributions.

(2) Lines 54–61: The current summary of prior work on composite-letter DNA storage is incomplete and may inadvertently mislead readers about the actual performance trade-offs.

(i) Six-letter experiments (Σ_6). Anavy et al. (2019) demonstrated data recovery at 29× coverage using KL-divergence inference followed by Fountain and Reed–Solomon coding. More recently, Xu et al. (2024) showed that a soft-decision decoder (Derrick-cp) applied to the same six-letter alphabet reduced the practical coverage threshold to 17×. Including this update will provide a realistic comparison point.

(ii) Large alphabets (256 letters). The early estimates of 2000× coverage from Anavy et al. (2019)’s simulations with a 256-letter composite alphabet stemmed from high-density aims (~8 bits/letter). Xu et al. (2024)’s Derrick-cp soft-decision decoding algorithm reduced requirements to 490×. Importantly, these elevated coverage needs are not signs of irrational design, but rather inherent consequences of pursuing high-density information encoding. The key question is whether advanced strategies (KL-divergence, soft-decision, or the present set-partitioning) can offset this cost effectively.

(3) Lines 80-82: The current Fig. 1f and Extended Data Table 1 inaccurately report Xu et al.’s metrics: Their six-letter performance derived from in vitro experiments following Anavy et al., not simulations; The value of 2.06 bits/cycle includes primer and index regions, while Extended Data Table 1 consistently omits these overheads. Excluding primers/indexes, the correct density is 2.39 bits/cycle. The revised methods and corresponding numerical values are listed in bellowed table. Therefore, in direct comparison with Xu et al.—considering both logical density and coverage—the performance of the current eight-letter system is comparable, but does not demonstrate notable advantages. I recommend updating Fig. 1f and Extended Data Table 1 to correct these values and revising the accompanying text to reflect that actual performance.

2. The main motivation for composite alphabets is to reduce synthesis cycles, which dominate the costs in DNA Storage system. However, the current manuscript does not quantify this critical trade-off. A comparative cost analysis aligned with prior studies would clarify the practical benefits of the proposed scheme.

3. Supplementary Tables 6–8 show that set-partitioning (SP) yields much lower error rates for standard bases (A/T/C/G) but higher error rates for composite letters (R, Y, M, K, etc.), which dominate the overall error budget, compared to KL-divergence (e.g. at 15×, error ≈4.8% for composites under SP vs 4.2% under KL). The authors should conduct end-to-end decoding performance comparison, using the same outer error-correction scheme (e.g., Reed-Solomon or LDPC), with detection via either set-partitioning or KL-divergence. This will clarify real gains (if any) in net data recovery. Furthermore, Xu et al. (2024) demonstrated that a MAP-based decoder can achieve >99% letter accuracy at 19× coverage—and still 98.3% accuracy even at 15×—clearly outperforming SP (demonstrated in Fig 5d and Supplementary Tables 6–8). The authors should therefore include MAP in their performance comparison to elucidate SP’s relative strengths and limitations.

4. While the conceptual use of set-partitioning is sound, its current entropy-based implementation appears suboptimal for composite letters. The results suggest the proposed method reduces within-subset confusion at the expense of mis-grouping between subsets. The authors appear to have directly ported the telecommunications partitioning approach without adaptation to composite letter characteristics. I suggest to develop a more tailored partitioning approach to accurately separate subsets.

5. The manuscript reports discarding ~23% of reads during length filtering (Fig. 5g), a shocking large fraction for Illumina sequencing (which typically has very low indel rates). The authors should explain the source of these indels and verify that filtering did not bias the results. Additionally, it is unclear whether the stated coverage thresholds (e.g. 18× for error-free recovery and in Fig. 5e-f) refer to pre- or post-filter read counts. Since coverage directly translates to DNA-storage readout cost, it should be defined on the basis of raw sequencing depth rather than the subset surviving filtering.

6. The manuscript lacks critical detail regarding the synthesis methodology for composite letters, which is a key practical challenge in composite DNA storage. In particular, it is not clear whether each oligo encoding a composite letter is

individually designed (deterministic assignment of A/T/C/G at each oligo) or whether the synthesis process uses probabilistic base mixing at degenerate positions, resulting in random incorporation across molecules. This distinction impacts scalability, and decoding accuracy. Also, discuss scalability implications for large-scale oligo pools.

Minor reversions

Line 141: The Galois field definition for NB-LDPC code appears incorrect—please correct the GF notation.

(Remarks on code availability)

Version 1:

Reviewer comments:

Reviewer #1

(Remarks to the Author)

I appreciate the authors for their detailed responses, which have addressed most of my earlier concerns. However, one remaining issue relates to the statement in the revised manuscript that “homopolymer length distributions in composite systems are comparable to conventional four-letter storage.” In fact, conventional DNA data storage systems do not generally use unconstrained four-letter encoding. A widely adopted strategy (Goldman et al., Nature, 2013) is to convert the binary data to base-3 digits (trits), and then convert to DNA code by replacement of each trit with one of the three nucleotides different from the previous one used. This strategy explicitly prevents homopolymers in every DNA strand. Thus, the authors may clarify that conventional systems can avoid homopolymers through appropriate encoding algorithms rather than simply employing 4-letter representations. More details can be found in this paper:

Goldman, N., Bertone, P., Chen, S. et al. Towards practical, high-capacity, low-maintenance information storage in synthesized DNA. Nature 494, 77–80 (2013).

(Remarks on code availability)

Reviewer #2

(Remarks to the Author)

I thank the authors for their considerable efforts in revising this manuscript. The manuscript is strengthened by the inclusion of 15-letter in vitro experiments, data written through two composite-letter synthesis platforms tailored to different oligo scales, and additional degradation experiments assessing DNA stability. With Illumina sequencing for readout, all decoding experiments achieved 100% data recovery, demonstrating the robustness of the proposed framework. Overall, the concept of “DNA diamond” is original and appealing, offering a useful entropy-based perspective on composite-letter codec for DNA storage.

For further clarification and fairness in reporting, I have a few minor suggestions:

1. When comparing overall performance (lines 86–100), please pair sequencing coverage with a net information density that includes all synthesized nucleotides (including primers and indices), or at least add the index overhead. In this design, the double-ended indices are intentionally introduced to mitigate strand displacement/oligo breaks and degradation (lines 301–302) and therefore form part of the error-correction mechanism that enables lower recovery depth; omitting them can overstate payload efficiency. By contrast, in prior studies such as Xu et al. and Anavy et al., indices primarily serve addressing, so payload-only metric would be reasonable within those contexts. The cost analysis should follow the consistent accounting and incorporate index costs to ensure fair comparison.

2. When positioning against prior studies (lines 351–354), please compare like-for-like alphabet sizes (e.g., 15-letter vs earlier 15-letter systems; for 8-letter vs 6-letter.).

3. The statements near lines 99–101 about advantages of the 15-letter system should be calibrated to match the evidence shown in Fig. 1f and Supplementary Table 10, where—under comparable 15-letter conditions—the storage density appears lower than that in Anavy et al.

(Remarks on code availability)

Version 2:

Reviewer comments:

Reviewer #2

(Remarks to the Author)

I appreciate the authors' detailed response and the rigorous revisions made to the manuscript, which have adequately addressed my previous concerns. Before publication, I have just two minor suggestions to harmonize the text with the supplementary data.

1. Lines 93–97: The revised text reports a density of 1.51 bits/letter for "payload and indices", which appears to reference the net information density (including primers). According to Supplementary Tables 1 and 4, the correct logical density for "payload and indices" is 2.23 bits/letter. Please correct the main text to align with your tabular data.

2. Lines 22–25 and 379–381: When highlighting the system's overall performance, pairing the sequencing coverage with the logical density of "payload and indices" (e.g., 33× sequencing coverage with 2.23 bits/letter) is more appropriate. This ensures the reader fully appreciates the efficiency of the error-correction mechanism (the indices) that enables such low recovery depth.

(Remarks on code availability)

Response to the Reviewers' Comments

We sincerely thank both reviewers for their insightful comments on our manuscript and greatly appreciate their constructive remarks and suggestions. We have revised the manuscript accordingly based on their valuable feedback and provide detailed point-by-point responses below. All changes in the revised manuscript are highlighted in yellow.

Response to the Reviewers #1

This work discussed an interesting approach, named DNA diamond, which is a composite letter constellation model for encoding in DNA data storage. This model expands the letters available for DNA data storage from 4 (simply natural bases A, T, C, G) to 15, considering the probability distribution of natural bases. Compared with other composite letter approaches, this method achieved the highest bit-to-letter coding efficiency with a relatively low sequencing coverage required to fully recover the data. The authors also developed methods to enable accurate and lossless recovery of data, including set partitioning, maximum likelihood estimation (MLE), appending double-end indices and perform length filtering.

The concept of DNA diamond is great. However, in practice, one of the key challenges of DNA data storage is the synthesis cost, which is measured by how many bits are encoded in one nucleotide. Although this work did a great job in improving the letter efficiency, I am not able to find any elaborations on the actual coding potential of this coding strategy. How many nucleotides, in total, are synthesized to encode each experimental (7,560-bit and 18,900-bit data) and simulation (780,713-byte image)? In my understanding, as shown in Supplementary Figure 2, N strands of 60-nt DNA only yielded 60 letters. If 1 letter can encode 3 bits in their wet experiments, then these strands just encoded 180 bits of data. I suggest the authors to calculate and clearly state the coding potential (not considering the indices, PCR primers and error correction codes) and the net information density (all synthesized nucleotides considered) using DNA diamond, with a unit of “bits/nt”.

Response: Thank you for your insightful comments. We are sorry for this confusion. We provide a detailed clarification below.

(1) Stored data volume and encoded strands (Supplementary Table 4)

In our prior small-scale experiments (before this revision), 7,560 bits and 18,900 bits were encoded into 7,560 composite letters, which were partitioned into short fragments of 60 letters, yielding a total of 126 composite strands (Supplementary Fig. 2). We

compared different eight-letter alphabets using four experiments. Therefore, 30,240 composite letters were synthesized and tested in all four experiments.

During manuscript revision, we further added validation on large-scale oligonucleotide pools. In our new experiments, two image files of 187,500 bytes and 234,375 bytes were encoded into 600,000 composite letters, which were then partitioned into 10,000 strands of 60 letters (Supplementary Fig. 3). We tested two different index schemes. Therefore, 2,400,000 composite letters were synthesized and tested during manuscript revision, with 1,200,000 using an eight-letter alphabet and 1,200,000 using a 15-letter alphabet.

The comparison with existing works can be found in the updated Supplementary Table 1. A summary can be found in Table R1 below.

Table R1. Data volume in ‘wet’ experiments (practical synthesis & sequencing)

Scheme	Data volume	File type	Pool
Small-scale verification	945 bytes	 Tang poems	126 composite strands (116 letters)
	2,355 bytes	 Tang poems	126 composite strands (116 letters)
	945 bytes	 Tang poems	126 composite strands (116 letters)
	945 bytes	 Tang poems	126 composite strands (116 letters)

Large-scale verification	187,500 Bytes	House.jpg (2,209 × 2,206 pixels)	10,000 composite strands (124 letters)
	187,500 Bytes	House.jpg (2,209 × 2,206 pixels)	10,000 composite strands (112 letters)
	234,375 Bytes	Female.jpg (3,045 × 3,045 pixels)	10,000 composite strands (124 letters)
	234,375 Bytes	Female.jpg (3,045 × 3,045 pixels)	10,000 composite strands (112 letters)

Total	829 KB		40,504 composite strands

(2) Workflow of composite DNA storage (How many nucleotides, in total, are synthesized?)

In order to clarify the data volume and synthesized nucleotides, we illustrate the basic workflow of our scheme with an eight-letter composite DNA storage process as an example (Supplementary Figs. 4 and 5).

In our scheme, digital data were first converted into binary bits. Every three bits were then mapped to a single composite DNA letter. For example, in the mapping table, ‘000’ corresponds to ‘A’, ‘001’ to ‘T’, ‘010’ to ‘G’, etc. Using this rule, 22,680 bits of data were converted into 7,560 composite letters. Next, these letters were divided into 126 strands, each strand carrying 60 composite letters (Supplementary Fig. 5).

During synthesis, each composite letter was realized as a programmed mixture of the four natural bases (A, T, G, C). For instance, the composite letter ‘Y’ represents a 50% mixture of C and T, while ‘M’ represents a 50% mixture of A and C. At each synthesis cycle, a large number of ordinary DNA strands (Supplementary Table 3) were added a base in parallel within a single CPG column or a single synthesis well (Supplementary Fig. 4). Each CPG column or synthesis well was corresponded to a single composite strand. For column-based synthesis, we used 126 columns in each experiment. We performed four experiments. For array-based inkjet printing synthesis, we used 10,000 wells to generate 10,000 composite strands. We also performed four experiments. Two used Alphabet 1 (8 letters), and the other two used Alphabet 4 (15 letters).

As shown in Supplementary Fig. 8 (Supplementary Fig. 2 in previous version), during data readout, many noisy ordinary reads are collected for each composite strand. A total number of n sequencing reads corresponding to a unique index were used to reconstruct the original composite strand.

Therefore, when we synthesis only a single composite strand, we practically synthesized a very large number of ordinary DNA strands (real oligos). When we read out a single composite strand, we require to read out n ordinary DNA strands (real oligos) to get n sequencing reads for recovery of this specific composite strand.

(3) Theoretical coding potential and the net information density

In four-base DNA storage systems, information density is usually expressed in bits per nucleotide (bits/nt). In composite-letter systems, each synthesis cycle produces one composite letter that represents a mixture of multiple nucleotides. Therefore, bits/letter may be more proper to describe the information density of composite alphabets^{R1, R2}.

- For the eight-letter system, the theoretical coding potential is $\log_2(8) = 3$ bits/letter. In our column-based synthesis experiment, 18,900 bits were stored in 7,560 letters, giving a payload logical density of 2.5 bits/letter (using RS(1890, 1575) code), excluding indices and primers. The logical density including indices and primers is 1.29 bits/letter. In our array-based DNA synthesis experiments, 187,500 bytes were stored in 600,000 composite letters. The payload logical density is 2.5 bits/letter. The logical density including indices and primers is 1.21 bits/letter (strand length: 124 letters) and 1.34 bits/letter (strand length: 112 letters), respectively.
- For the 15-letter system, the theoretical coding potential increases to $\log_2(15) \approx 3.9$ bits/letter. We achieved a density of 3.125 bits/letter in the payload segment (due to ECC), excluding indices and primers. The logical density including indices and primers is 1.51 bits/letter (strand length: 124 letters) and 1.67 bits/letter (strand length: 112 letters).

Details are presented in Supplementary Table 4.

References

- [R1]. Anavy, L., Vaknin, I., Atar, O., Amit, R. & Yakhini, Z. Data storage in DNA with fewer synthesis cycles using composite DNA letters. *Nat. Biotechnol.* **37**, 1229–1236 (2019).
- [R2]. Xu, Y., Ding, L., Wu, S. & Ruan, J. Overcoming the high error rate of composite DNA letters-based digital storage through soft-decision decoding. *Adv. Sci.* **11**, 2402951 (2024).

We revised main manuscript and supplementary information as follows.

Lines 83–86:

To verify the proposed scheme, we designed error correction codes tailored to different alphabets and conducted experiments on both column-based and array-based synthesis platforms, along with simulation tests (Supplementary Figs. 2, 3 and 4).

Lines 152–166:

Then, we validated the applicability of discrete-entropy composite letters by testing multiple storage schemes across different codes and alphabets (Fig. 2c, Supplementary Figs. 10 and 11, Supplementary Table 4). The encoding and writing process of the composite DNA storage system included three stages: data encoding, composite letter mapping, and DNA synthesis.

For data encoding, in the eight-letter system, 945 bytes of text were first encoded using NB-LDPC(3780,1260) code ($R=0.33$) over Galois Field ($GF(64)$), with interleaving to mitigate burst erasures³⁹ (Supplementary Fig. 12). To validate high logical density, 2,355 bytes of text data were encoded with RS(1890,1575) code ($R=0.83$) over $GF(2^{12})$, achieving 2.5 bits/letter (Supplementary Fig. 13). We further extended the eight-letter scheme to large-scale pools, where a 187,500-byte image file was encoded using RS(3750,3125) code (Supplementary Fig. 14). In the 15-letter system, both experiments and simulations were performed. A 234,375-byte image file was encoded with RS(30000,25000) code ($R=0.83$, 3.125 bits/letter) for experimental verification (Supplementary Fig. 15). A 780,713-byte image file was encoded with RS(32767,27767) code ($R=0.85$, 3.18 bits/letter) for simulations (Supplementary Note 2, Supplementary Fig. 16).

In Supplementary information:

- (1) **Supplementary Table 4** is added to provide detailed information on the amount of stored data, the number of synthesized composite letters, and the information density.
- (2) **Supplementary Note 2** is added to provide the logical density computation of the composite letter DNA storage.
- (3) **Supplementary Figs. 2–5** are added to illustrate our experimental and simulation setups as well as the workflow of composite letter DNA storage.

Supplementary Table 4. Coding potential and the net information density.

Scheme	Column-based DNA synthesis experiments		Array-based DNA synthesis experiments				Simulation
Data volume (bytes)	945	2,355	187,500	234,375	234,375	780,713	
Bit information (bits)	7,560	18,900	1,500,000	1,875,000	1,875,000	6,247,575	
Payload (M*L)	126*60 =7560 letters	126*60 =7560 letters	10000*60 =600,000 letters	10000*60 =600,000 letters	10000*60 =600,000 letters	32767*60 =1,966,020 letters	
Alphabet	Σ_8	Σ_8	Σ_8	Σ_8	Σ_{15}	Σ_{15}	
Coding potential (bits/letter)	$\log_2(8) = 3$	$\log_2(8) = 3$	$\log_2(8) = 3$	$\log_2(8) = 3$	$\log_2(15) = 3.9$	$\log_2(15) = 3.9$	
Coding scheme	NB-LDPC(3780, 1260)	RS(1890, 1575)	RS(3750, 3125)	RS(3750, 3125)	RS(30000, 25000)	RS(32767, 27767)	
Code rate	$R=1/3$	$R=5/6$	$R=5/6$	$R=5/6$	$R=5/6$	$R=17/20$	
Logical density (payload) (bits/letter)	1 bit/letter	2.5 bits/letter	2.5 bits/letter	2.5 bits/letter	3.125 bits/letter	3.18 bits/letter	
Synthesized nucleotides (including primers/indices)	126*116 =14,616	126*116 =14,616	10000*124 =1,240,000	10000*112 =1,120,000	10000*124 =1,240,000	10000*112 =1,120,000	/
Net information density (synthesized nucleotides) (bits/letter)	0.52 bits/letter	1.29 bits/letter	1.21 bits/letter	1.34 bits/letter	1.51 bits/letter	1.67 bits/letter	/

Supplementary Figure 2. Experimental verification using column-based synthesis and simulation. **a**, Four independent experiments were performed using column-based phosphoramidite chemistry. Two code rates, 1/3 and 5/6, were employed in combination with three distinct eight-letter alphabet schemes. In each experiment, 126 designed sequences with a payload of 60 composite letters and a total length of 116 nucleotides were synthesized. Each of the 126 sequences was individually synthesized in a separate column well. **b**, Computer simulation using the USC-SIPI image dataset (780,713 bytes). Data were encoded with an RS code ($R=0.847$), mapped to the 15-letter alphabet (15-bit to 4-letter) and formed 32,767 composite payload strands of 60 letters.

Supplementary Figure 3. Experimental verification using array-based DNA synthesis. **a**, Two composite oligo pools based on the eight-letter alphabet were synthesized using low-cost array-based inkjet phosphoramidite chemistry. Each experiment encoded a 187,500-byte image file into 600,000 composite letters, segmented into 10,000 strands of 60 letters. After adding 20-nt primers and indices of 12 nt or 6 nt, the resulting oligonucleotides were 124 letters and 112 letters, respectively. **b**, Two composite oligo pools based on the 15-letter alphabet were synthesized. Each experiment encoded a 234,375-byte image file into 600,000 composite letters, segmented into 10,000 strands of 60 letters. All four pools were synthesized using array-based inkjet DNA synthesis.

Supplementary Figure 4. Illustration of the composite strand synthesis with column-based and array-based synthesizers. **a**, Column-based synthesis of composite DNA strands. Eight reagent channels correspond to eight composite bases. Three composite alphabets were chosen and 126 composite strands were synthesized for each alphabet. **b**, Each designed composite strand, composed of composite letters (e.g., B, G, T, D, A, etc.), is individually synthesized in a dedicated well using column-based DNA synthesis. **c**, Array-based inkjet synthesis of large-scale composite-letter pools. Four printheads deliver monomer droplets onto array chips following predefined base mixing ratios.

Supplementary Figure 5. Writing workflow of composite DNA data storage. First, the encoded bits are mapped to composite DNA letters (e.g., mapping 3 bits to 1 letter), where each composite letter represents a predefined mixture of nucleotides (e.g., A, T, G, C, R, Y, M, K). Second, 126 composite strands with a payload length of 60 letters are generated. Third, these sequences are synthesized in parallel using column-based phosphoramidite chemistry, in which different nucleotide mixtures are deposited in each synthesis cycle. Finally, ordinary DNA strands are produced, forming clusters that reflect the designed composite letters.

Except the major concern above, I also have other concerns in terms of the feasibility of this coding strategy in real settings:

1. In practice, the existence of homopolymers (3 or more) of natural DNA bases is detrimental to sequencing-based data recovery. However, by mapping 1 letter to 3-bit of data, it is impossible to avoid the appearance of homopolymers in synthesized DNA strands. How would the authors overcome this issue?

Response: Thank you for raising this important point about the homopolymers.

We agree with the reviewer that it is impossible to avoid the appearance of homopolymers in synthesized DNA strands. To evaluate this concern, we performed a detailed analysis of the homopolymer length distribution with raw sequencing data.

Homopolymers are present in both conventional and composite letter storage systems. We analyzed the **proportion of homopolymers**. The sequencing reads from our composite strands show nearly identical distributions to those of randomly generated DNA sequences (**Supplementary Fig. 54**).

The probability of long homopolymers decreases sharply as the homopolymer length increases. Therefore, we did not control the homopolymer length dedicatedly. We exploited the low probability to avoid long homopolymers. This method has also been verified in [R1].

Then, we used two schemes to overcome the potential effects due to homopolymers.

(1) **Frequency-based composite letter detection alleviates the effects of homopolymers.** Composite letter readout depends on measuring base frequency distributions at each synthesis site (**Supplementary Fig. 55**). Even when sequencing errors (e.g., indels due to homopolymers) occur within homopolymer regions, with multiple copies available, the model infers each composite letter from the observed base frequencies across all aligned reads. The effects of an error in a single sequencing read (including indels due to homopolymer) can be further reduced.

(2) **Length filtering.** When indel errors occur within homopolymer regions and cause local shifts, misaligned bases can distort frequency observations and reduce detection accuracy. Our workflow filters out reads with abnormal lengths to further suppress indel noise (**Supplementary Fig. 58**). As detailed in Section 5 of the manuscript, length filtering improves per-letter detection accuracy and lowers the sequencing coverage required for error-free recovery (**Fig. 5e, f**).

The revised main text in the manuscript is as follows:

Lines 296–300:

Notably, homopolymers are unavoidable in sequencing reads and long homopolymers may introduce more indels. Fortunately, long homopolymers occurred rarely.

Homopolymer length distributions in composite systems are comparable to conventional four-letter storage and random sequences (Supplementary Figs. 54 and 55).

In the **Supplementary information**:

Supplementary Figs. 54, 55, and 58 are added to show homopolymer occurrence in composite letter storage system and how frequency-based inference combined with length filtering reduces their impact on letter detection.

[R1] Weindel, F., Gimpel, A. L., Grass, R. N. & Heckel, R. Embracing errors is more effective than avoiding them through constrained coding for DNA data storage. in *2023 59th Annual Allerton Conference on Communication, Control, and Computing (Allerton)* 1–8 (2023). doi:10.1109/Allerton58177.2023.10313494.

a**Sequencing reads from composite letter strand (measured in bases)****b****Reads from Fountain coding (measured in bases)****c****Randomly generated sequence (Measured in bases)**
Supplementary Figure 54. Homopolymer length distribution. **a**, Sequencing reads from composite letter strands (measured in bases). The proportions of single-base homopolymers are 75.84% in the 8-letter system and 75.46% in the 15-letter system. **b**, Sequencing reads from a Fountain coding–based natural DNA system (measured in bases), with 75.90% single-base homopolymers. **c**, Randomly generated sequences. The proportion of single-base homopolymers is 75.39% in the natural-DNA system (measured in bases).

Supplementary Figure 55. Frequency-based inference of composite letters. **a**, Multiple sequencing reads with the same indices are aligned, and the observed base frequencies at each synthesis site are used to infer the most probable composite letter. **b**, Indel errors in homopolymer regions cause subsequent bases to misalign. **c**, Composite letters are detected based on frequency distributions from multiple copies, which offset the effect of local errors.

Supplementary Figure 58. Readout workflow with length filtering. Sequencing reads are first aligned to double-end primers to identify payload and index segments (Primer alignment). Reads with abnormal length are discarded, ensuring equal-length, aligned sequences for downstream readout (Length filtering). Reads are then grouped into different clusters according to the index sequences (Index identification). Within each cluster, base frequency distributions at each position are calculated across multiple copies, and the most probable composite letter is inferred (Letter detection).

2. In their simulation experiment, the mapping between 15 binary bits ($2^{15} = 32,768$) and 4 letters ($15^4 = 50625$) left so many combinations of letters not used in encoding and therefore is not efficient. This originates from the fact that this encoding strategy is very sensitive to the total bits of information to be coded, and the coding efficiency is not always high for different data sizes. In addition, I noticed that for different sizes of data, a different size of block should be carefully selected (6-bit in wet experiment and 15-bit in simulation). Did the authors investigate the impact of the size of block on coding potential?

Response: Thank you for this insightful question.

When mapping 15 bits to 4 letters of the 15-letter alphabet, not all possible letter combinations are used, which inevitably leads to some redundancy. The theoretical density of the 15-letter alphabet is $\log_2(15) \approx 3.9069$, which is not an integer. Each letter can maximally store about 3.9069 bits.

When grouping 4 letters together, the storage capacity is $4 \times 3.9069 \approx 15.6$ bits, slightly higher than 15 bits and lower than 16 bits. Therefore, if we choose 15 bits as a group, it can be represented by 4 letters, with an efficiency of $15/15.6 \approx 0.96$.

If 16 bits are chosen as one block, 5 letters are required, and $5 \times 3.9069 \approx 19.5$ bits, which is greater than 16 bits. However, $2^{16} = 65,536$, while the total number of combinations of 5 letters is $15^5 = 759,375$, leading to lower efficiency.

Therefore, in our storage system, mapping efficiency is used as a key metric to evaluate encoding efficiency. The detailed calculation is as follows.

(1) Mapping efficiency of 15-letter and eight-letter systems

For the 15-letter alphabet, the theoretical density is $\log_2(15) \approx 3.9069$ bits per letter. When the block size is set to 15, the required number of letters is $15/\log_2(15) \approx 3.84$, rounded up to 4 letters. The mapping efficiency is defined as the ratio of stored binary bits to the maximum capacity of the letters used. Therefore, the efficiency of 15 binary bits is $15/(4 \times \log_2(15)) \approx 0.96$ (**Supplementary Fig. 17**).

For the eight-letter alphabet, any 3 bits can be directly mapped to one letter. Given $\log_2(8) = 3$ bits/letter, the mapping efficiency is 100%. In our current composite storage system, limited by synthesis feasibility, we validated both the 15-letter and eight-letter, as these represent the practical design choices.

(2) Relationship between mapping efficiency and bit group size

In our design, we presented a proper and high-efficiency scheme. We are sorry that

we did not present the detailed explanation on this point. In this revised version, we have added the details on how to choose the conversion block size.

The mapping efficiency depends on both the chosen block size and the alphabet size. We derived a formula to calculate mapping efficiency as a function of block size, which can be expressed as:

$$\eta = \frac{\log_2(2) \times n}{(\log_2(k) \times \lceil n / \log_2(k) \rceil)},$$

where n is the size of the binary data block (in bits), k is the size of the composite alphabet (e.g., $k = 8$ or $k = 15$), and $\lceil \cdot \rceil$ denotes the ceiling function to calculate the number of composite letters required to represent n bits of data. The efficiency fluctuates with block size (**Supplementary Fig. 18**). When the block size is set to $2^N - 1$ (e.g., $n = 31$), the efficiency is relatively high (≈ 0.99), but each block requires multiple composite letters (e.g., it is eight for $n = 31$). In this case, a single error in one letter would corrupt many bits in the 31-bit block, causing error propagation across all the bit group (e.g., $n = 31$). To balance efficiency and error control, we selected a block size of 15 bits ($n = 15$), which achieves an efficiency of 0.96 while limiting error propagation to within 15 bits.

Furthermore, we chose $n=15$, also because it is quite compatible with our RS codes defined in $GF(2^{15})$. For our RS code, a single RS symbol also contains 15 bits.

We added **Supplementary Figs. 17 and 18** to illustrate the mapping process of the bit-to-letter. We also added **Supplementary Note 3** to further clarify the mapping efficiency between binary bits and composite letters (please find in the supplementary information).

The revised text has been added in the manuscript:

Lines 167–173:

For composite letter mapping, we chose four composite letter alphabets (Fig. 2d) and analyzed the impact of different bit group sizes on mapping efficiency (Supplementary Figs. 17 and 18). The eight-letter alphabet was implemented by converting every three bits into a single composite letter. The mapping efficiency is 100%. For the 15-letter alphabet, a group of 15 bits was converted into four composite letters, corresponding to an optimized efficiency of 96%. We chose the group size of 15 for satisfactory efficiency and error propagation within the group. Details can be found in Supplementary Note 3.

In the **Supplementary information**:

(1) **Supplementary Figs. 17, and 18** are added to illustrate the mapping process of the bit-to-letter.

(2) **Supplementary Note 3** is added to provide a textual explanation of the mapping efficiency between binary bits and composite letters.

Supplementary Figure 17. Mapping efficiency of bits to composite letters. Binary blocks of n bits are mapped to composite letters from an alphabet of size k , requiring $\lceil n / \log_2(k) \rceil$ letters. Examples show that for an eight-letter alphabet, 3 bits can be perfectly represented by a single composite letter, achieving a mapping efficiency of 100%. In contrast, for a 15-letter alphabet, 15 bits require 4 letters, and the mapping efficiency is 96%.

Supplementary Figure 18. Bit-to-alphabet mapping efficiency as a function of bit group size. **a**, Mapping efficiency from binary bits to the eight-letter alphabets. **b**, Mapping efficiency from binary bits to 15-letter alphabets. Mapping efficiency is defined as the ratio between the number of input bits in each block and the amount of information that can be represented using a fixed number of multi-valued symbols. The red stars indicate the configurations used in our study. Every three bits are mapped to a single octal symbol in the “wet” experiment, and a 15-bit group is mapped to four base-15 symbols. For the 15-letter system, this mapping achieves a mapping efficiency of 96%. **c**, Example of letter mapping and de-mapping in the 15-letter composite encoding scheme. During sequencing or synthesis, a letter error may occur (for example, “H” is misread as “V”), leading to an incorrect symbol after de-mapping. Each 15-bit symbol is protected by Reed–Solomon coding (RS(30000, 25000) over $GF(2^{15})$), enabling correction of such symbol-level errors during decoding.

3. I am not clear on how the authors can accurately control the read count of each DNA strand through sequencing. I assume this was achieved through adjustment of copy numbers of each strand. However, this significantly complicates experimental procedure and is apparently not practical in commercial DNA data storage settings. In addition, the efficiency of PCR is different for different amplicon sequences (the payload in this study). For example, the amplification may be difficult to proceed when there are homopolymers in the amplicon. When this encoding strategy scale up to 10,000 or more strands in one oligo pool (which is very typical in real DNA data storage applications), can the probability distribution of each natural DNA base still be guaranteed?

Response: We thank the reviewer for this important question. In our study, we did not attempt to control the read count of each DNA strand during sequencing. Instead, we relied on pooled DNA manipulation and statistical sampling.

Specifically, we performed downsampling of raw sequencing reads on pooled all DNA strands and evaluated the recovery performance at different average coverages. To avoid ambiguity, we revised all mentions of “coverage” in the manuscript to “average coverage”.

(1) Experimental validation of small-scale oligo pools (126 composite oligos).

In Part 4 of the Results, we analyzed strand copy number variation across different average coverages, with a particular focus on low-coverage conditions.

All composite oligos were pooled together for amplification and sequencing. Therefore, we did not control the copies of each composite strand. Actually, we could not control the copies. Due to stochastic sampling, bias in PCR amplification efficiency and sequencing across DNA strands, the observed read counts were found to follow a negative binomial distribution (**Supplementary Figs. 29 and 30**). The probability distribution of natural bases at composite positions varies across strands and is subject to random sampling noise as well as synthesis, amplification, and sequencing stochasticity. At lower sequencing coverages, this manifests more clearly in the form of dropout (i.e., erasures) or uneven base frequencies (**Figure 4a and 4b**). Despite this, most composite letters maintain the desired probability distribution of each natural DNA base (**Supplementary Figs. 19–21**) and also maintain separable entropy characteristic (**Fig. 4a, b**).

Furthermore, our decoding strategy incorporates both erasure correction and substitution recovery, enabling accurate data readout even under low-coverage conditions (**Supplementary Figs. 42 and 43**).

(2) Experimental validation of large-scale oligo pools (10,000 composite oligos).

In response to the reviewer’s suggestion, we further validated our encoding scheme in large-scale oligo pools, considering that large pools show significant bias.

Specifically, we designed four oligo pools. Each contained 10,000 composite strands

based on the eight-letter or 15-letter alphabets (Supplementary Figs. 14 and 15). The pools were synthesized using a high-throughput inkjet-based oligonucleotide synthesizer and then sequenced with NGS. We evaluated read count distributions, letter detection error rates, and recovery performance under different average coverages. The results showed that even in pools containing 10,000 oligonucleotides, the observed read counts followed a negative binomial distribution (Supplementary Figs. 31–33). When the composite-letter DNA storage system is scaled to a pool of 10,000 strands, the probability distribution for each composite letter remains valid and retain entropy discriminability (Supplementary Figs. 22–26).

With increasing coverage, composite letters can be reliably inferred, and the encoded data can be recovered (Supplementary Figs. 44–47).

We updated Part 4 of the Results. The revised manuscript is as follows:

Lines 211–217:

The coverage parameter in the model was obtained by fitting experimental copy number distributions at different average sequencing depths. For column-based synthesis pools, the observed read counts per strand followed a negative binomial distribution (Supplementary Figs. 29 and 30). Consistently, in array-based synthesis pools containing up to 10,000 oligonucleotides, the read count distributions also conformed to the negative binomial model (Supplementary Figs. 31–33). This reflects stochastic sampling and strand-specific differences in PCR amplification efficiency.

In the Supplementary information:

Supplementary Figs. 25–28 are added to present read-count distributions under different average sequencing coverages, highlighting non-uniform strand representation and extending the analysis to 10,000-strand pools for different alphabets.

Supplementary Figure 30. Read count distribution of composite DNA strands at different average coverages (126 composite strands). **a**, Distribution of sequencing read counts per composite strand synthesized by the column-based method (NB-LDPC(3780, 1260) code with an eight-letter alphabet). The left panel shows that the read count distributions at different coverages follow a negative binomial distribution. The parameter p denotes the fitted success probability of the negative binomial model. The right panel shows the read count distribution across 126 composite strands at 9× average coverage, where error-free data recovery was achieved. The read counts exhibited noticeable non-uniformity. **b**, Distribution of sequencing read counts from the composite pool encoded with the RS(1890, 1575) code using the eight-letter alphabet. The left panel shows the fitted negative binomial distributions at different coverages. The right panel presents the read-count distribution of individual composite strands (126 in total) at 14× average coverage, where error-free recovery was obtained at this coverage.

Supplementary Figure 31. Read count distributions at different sequencing coverages (10,000 composite strands). **a**, Read count distribution of 10,000 composite strands with an eight-letter alphabet (array-based inkjet synthesis). The observed distributions at different average coverages follow negative binomial statistics. **b**, Distribution of read counts for 10,000 composite strands (124 letter length, 15-letter alphabet). In all cases, PCR amplification bias and random sequencing sampling produced negative binomial-like strand copy distributions, and the fitted parameter p quantifies the degree of non-uniformity across strands.

10,000 composite strands, 124 letters, Alphabet {A,T,G,C,R,Y,M,K}

Supplementary Figure 32. Read count distribution of 10,000 individual composite strands with a chosen eight-letter alphabet. Read count distribution of 10,000 composite strands obtained from the pool experiment using the eight-letter composite alphabet at an average sequencing coverage of 19×. Each bar represents the number of sequencing reads corresponding to a single composite strand.

10,000 composite strands, 124 letters, Alphabet {A,T,G,C,R,Y,S,W,M,K,H,B,V,D,N}

Supplementary Figure 33. Read count distribution of 10,000 individual composite strands with a chosen 15-letter alphabet. The read count distribution of 10,000 composite strands synthesized with the 15-letter composite alphabet at an average sequencing coverage of 33 \times . The copy number of each strand is non-uniform after sequencing.

Supplementary Figure 22. Base frequency distribution (1,000×, 10,000 composite oligos, eight-letter alphabet). a–h, Distributions of four constituent bases (A, T, G, C) for each of the eight composite letters at 1,000× average coverage at each synthesis position. The results are derived from sequencing data of 10,000 composite strands encoded with an eight-letter alphabet {A, T, G, C, R, Y, M, K}.

Supplementary Figure 23. Base frequency distribution (19 \times , 10,000 composite oligos, eight-letter alphabet). a–h, Distributions of four constituent bases (A, T, G, C) for each of the eight composite letters at 19 \times average coverage, representing the minimum coverage at which error-free data recovery was achieved. The results are derived from sequencing data of 10,000 composite strands encoded with the eight-letter alphabet {A, T, G, C, R, Y, M, K}.

1000× NGS reads

Supplementary Figure 24. Base frequency distribution (1,000×, 10,000 composite oligos, 15-letter alphabet). a–o, Distributions of four constituent bases (A, T, G, C) for each of the 15 composite letters at 1,000× average coverage. The results are derived from sequencing data of 10,000 composite strands encoded with the 15-letter alphabet {A, T, G, C, R, Y, S, W, M, K, H, B, V, D, N}.

33× NGS reads

Supplementary Figure 25. Base frequency distribution (33×, 10,000 composite oligos, 15-letter alphabet). a–o, Distributions of four constituent bases (A, T, G, C) for each of the 15 composite letters at 33× average coverage, representing the minimum coverage at which error-free data recovery was achieved. The results are derived from raw sequencing data of 10,000 composite strands with the 15-letter alphabet {A, T, G, C, R, Y, S, W, M, K, H, B, V, D, N}.

Supplementary Figure 26. Entropy distribution under practical synthesis and sequencing. **a** and **b**, Entropy distributions of the 15-letter alphabet at high coverage (1,000×) and the minimum coverage for error-free decoding (33×). **c** and **d**, Entropy distributions of the eight-letter alphabet at high coverage (1,000×) and the minimum coverage for error-free decoding (19×). Each curve represents the entropy distribution for individual composite letters under noisy sequencing conditions.

4. The degradation of DNA is also a concern in practice, and the degradation rate of DNA strands with different sequences may also be non-homogeneous, which cause distortion of the assumed probability distribution of natural bases. I suggest the authors to perform accelerated aging experiments on their synthesized DNA pools (elevated temperatures, acidic and basic conditions, repeated freeze-thaw cycles) to investigate if this encoding strategy is less resistant to DNA degradations compared with conventional encoding strategies.

Response: We thank the reviewer for this valuable suggestion. To assess the stability and robustness of our composite DNA strands under degradation, we performed accelerated thermal aging experiments and analyzed the effect of degradation on both physical strand integrity and data retrieval accuracy. We added **Supplementary Figs. 60–66**.

(1) Double indices enabling readout under DNA degradation (Supplementary Fig. 60)

Our composite strand incorporates index and primer sequences at both ends of the payload. During sequencing, any read containing at least one intact primer–index pair can still be correctly clustered and aligned according to its positional index. When strand breakage occurs, the remaining fragments with a single-end primer are sufficient to reconstruct missing positions by aggregating multiple overlapping copies, allowing the original composite letters to be inferred. This design inherently mitigates the impact of physical strand degradation on data decoding.

(2) Accelerated aging experiments.

We conducted gradient thermal degradation experiments on the small-scale pool, which consisting of 126 composite strands of 116 letters. Samples were incubated at 80 °C (2.5 h), 85 °C (2 h), 90 °C (2 h), and 95 °C (2 h), with an unheated control group. DNA breaks are induced frequently during DNA degradation, consequently leading to increased sequence dropout ^{R1, R2}.

(3) Sequencing and degradation analysis.

Each degraded sample was sequenced using the paired-end 150-bp mode on an NGS platform, and the sequencing reads were assembled from paired-end reads to reconstruct full-length fragments. The cumulative read-length distributions show that higher incubation temperatures lead to broader distributions, reflecting increased degradation (**Supplementary Fig. 61**). Quantitatively, the fraction of correctly assembled reads (116 nt) decreased with temperature, while truncated reads increased (**Supplementary Figs. 62 and 63**). Furthermore, primer-based classification of the reads showed that the proportion of reads containing both paired-end primers declined with increasing temperature. At 95 °C, approximately 14% of the reads carried only a single-end primer (**Supplementary Fig. 64**), which can be attributed to strand breakage.

(4) Data recovery and error analysis

We evaluated data readout performance using the degraded sequencing datasets. We added Supplementary Figs. 65 and 66 to illustrate the recovery performance. Although higher temperatures increased the inferred-letter errors, the overall recovery performance remained high (Supplementary Fig. 65). Specifically, the letter error rate rose moderately with temperature, but all degraded samples achieved complete data recovery. Furthermore, considering that the degradation was not severe, we performed a stress test using selected short degraded fragments. The original file was still recovered with a relatively high sequencing coverage (Supplementary Fig. 66).

These results demonstrate that our scheme is resilient to strand degradation and that increasing sequencing coverage effectively compensates for DNA damage.

(5) On the repeated freeze-thaw cycles

We have also performed repeated freeze-thaw cycles on synthesized DNA. However, we did not observe obvious degradation during limited freeze-thaw cycles (Figure R1). Considering that most of the degradations can be modeled as strand breaks^{R1, R2}, we used the gradient thermal degradation to verify our proposed scheme. Therefore, we expect the reviewer to allow us not to include these experiments in our modified manuscript.

Figure R1. Analysis of degradation due to repeated freeze-thaw cycles. a and c, Cumulative read-length distributions of assembled reads from two small-scale composite pools (126 strands, 116 letters) subjected to ten freeze–thaw cycles. Each cycle consisted of freezing at $-20\text{ }^{\circ}\text{C}$ for 30 min and thawing at room temperature for 6 min. The read lengths remained concentrated around the full 116 nt. **b and d,** Proportions of reads containing paired-end primers, only 5' primers, only 3' primers, or invalid primer structures. Over 96% of reads retained both primer sequences, confirming that repeated freeze–thaw treatment did not lead to strand breaks.

References

[R1]. Song, L. et al. Robust data storage in DNA by de Bruijn graph-based de novo strand assembly. *Nat. Commun.* **13**, 5361 (2022).

[R2]. Gimpel, A.L., Stark, W.J., Heckel, R. *et al.* A digital twin for DNA data storage based on comprehensive quantification of errors and biases. *Nat. Commun.* **14**, 6026 (2023).

Therefore, we added a new paragraph at the end of Results, Part 5 describing the resilience of composite DNA strands under thermal degradation, as detailed below.

Lines 312–326:

Then, the double-end strand structure enables robust data retrieval even when partial degradation occurs, for example, strand break. Any fragment retaining at least one intact primer–index pair can still be correctly clustered and aligned according to its index (Supplementary Fig. 60). To evaluate the performance on strand break, we performed accelerated thermal aging experiments. The composite pool containing 126 strands of 116 letters was incubated at 80 °C (2.5 h), 85 °C (2 h), 90 °C (2 h), and 95 °C (2 h), with an unheated sample as the control. Under higher temperatures, the read lengths showed a pronounced shift toward shorter fragments. Consistently, the proportion of correctly assembled 116 nt reads decreased, while truncated fragments accumulated (Supplementary Figs. 61–63). At 95 °C, nearly 14% of reads contained only a single-end primer (Supplementary Fig. 64). Despite these degradation, data recovery remained highly robust. Letter error rates increased moderately with degradation temperature. Error-free recovery was still achieved at sufficient sequencing coverage (Supplementary Fig. 65). Stress tests using short fragments revealed that the proposed scheme could recover the original error free, though the required sequencing coverage increased (Supplementary Fig. 66).

In the Supplementary information:

Supplementary Figs. 60–66 are added to present the composite DNA architecture, read length under strand degradation, data recovery performance, and stress tests using degraded reads.

Supplementary Figure 60. Design of composite DNA strand structure enabling degradation-tolerant decoding. **a**, Schematic of composite strand architecture with double-end primers and indices. Three sequencing scenarios are shown: (1) reads containing both paired-end primers, (2) reads containing only the 5' primer, and (3) reads containing only the 3' primer. **b**, Multiple sequencing reads sharing the same index are aggregated to compute position-wise observed frequencies and infer composite letters. Even if one end of the strand is degraded, the remaining indexed fragment contributes to composite letter detection.

Supplementary Figure 61. Cumulative distributions of read length under different degradation conditions. **a**, Cumulative frequency curves of assembled reads for the pool encoded with RS(1890, 1575). Increasing temperature led to a gradual shift toward shorter read lengths and a broader cumulative distribution, indicating progressive degradation of DNA strands. **b**, Magnified view of the 60–110 nt region. The original full-length reads were 116 nt in design, and the detailed view highlights the emergence of shorter degraded fragments at higher temperatures (95 °C, 2 h).

Supplementary Figure 62. Proportion of read length under different thermal degradation conditions (only payload). The y -axis is payload length (nt), and the x -axis is the proportion of reads when strands are sorted by payload length from long to short; i.e., for each proportion on the x -axis, the curve shows the corresponding payload length. Higher temperatures (95 °C, 2 h) shift the curves downward, indicating a larger fraction of shorter payloads.

Supplementary Figure 63. Magnified view of proportion of read length under different thermal degradation conditions (only payload). Detailed view of the payload ratio range 0.9–1.0. The payload exhibited a gradual shortening trend at higher temperatures. Degradation becomes more pronounced at elevated temperatures, with partial fractures observed within the payload region.

Supplementary Figure 64. Classification of sequencing reads according to primers under different thermal degradation conditions (126 composite strands, 116 nt). Proportions of reads containing paired-end primers, only 5' primer, only 3' primer, or invalid primers. The percentage of paired-end reads gradually decreases with increasing temperature, while single-end reads become more prevalent. At 95 °C, nearly 14% of reads exhibited fragmentation.

Supplementary Figure 65. Data recovery performance of degraded samples (126 composite strands, 116 nt). **a**, Relationship between valid base coverage and average coverage. Valid base coverage is defined as the ratio of the number of bases used for composite-letter detection to the total number of encoded letters. **b**, Index identification accuracy. The payload regions identified were used for composite-letter detection. **c**, Letter error rate as a function of average coverage for each condition (mean \pm s.d.). The non-heated control sample showed the lowest error rate. **d**, Erasure rate at different average coverages. **e**, Percentage of successful (error-free) recoveries across 100 independent trials using the RS(1890, 1575) coding scheme. **f**, Composition of valid payloads at 14 \times coverage by index identification, including double-end indices and those identified by only 5'-end or 3'-end indices.

Supplementary Figure 66. Recovery performance using severely-degraded sequencing reads. **a**, Length distribution of payload regions identified based on double-end indices. Approximately 40% of the fragments showed degradation within the payload region. **b**, Proportion of degraded reads containing double-end indices (38.71%), only 5'-end index (27.17%), or only 3'-end index (34.12%). **c**, Length distributions of payload regions identified by single-end indices, where sequences shorter than 60 nt indicate degradation within the payload region. **d** and **e**, Data recovery performance based on degraded reads, showing letter error rate and successful recovery rate as functions of average base coverage. **f**, Distribution of 491,147 degraded sequencing reads extracted from the aging experiment at 95 °C for 2 h.

Response to the Reviewers #2

This manuscript introduces a “diamond” composite-letter alphabet for DNA storage and an entropy-guided, two-stage detection pipeline: composite letters are first partitioned into entropy-defined subsets and subsequently identified within subsets by maximum-likelihood inference. Wet-lab experiments on an eight-letter alphabet report achieving ~2.5 bits per letter and error-free recovery at 18× Illumina coverage. Although these proof-of-concept results are promising, the study currently exhibits several empirical, methodological, and contextual shortcomings.

Response: We thank the reviewer for their detailed review of our manuscript. Below are our point-by-point responses and the modifications detailed in the manuscript. We expect the revised version should address the reviewer’s concerns.

1. The manuscript contains several instances of imprecise descriptions and inaccurate data. In particular, the authors should more accurately clarify the challenges of composite-letter storage and carefully contextualize prior studies.

(1) Lines 42-47: While the authors identify key manifestations of the composite-DNA readout challenge—namely, difficulty in discriminating similar base mixtures and obscuration of constituent components by sequencing noise—the discussion remains somewhat surface-level and does not sufficiently articulate the fundamental underlying causes. In fact, the elevated error rates in composite DNA storage arise from two interacting sources: (i) synthesis and sequencing errors, which perturb the intended base mixture proportions; and (ii) statistical sampling bias, especially under low coverage, which leads to fluctuations in observed base-frequency distributions.

Response: We thank the reviewer for this comment. In Part 4 of the *Results* (“Performance evaluation on constituent base frequency variation at low average coverage”), we analyzed how statistical sampling bias under low sequencing coverage affected base-frequency distributions. We quite agree with the reviewer.

We revised the relevant text in the manuscript as follows.

Lines 41–50:

Despite these advantages, composite DNA storage faces challenges during readout, particularly elevated error rates and the requirement of high sequencing coverage. These issues stem from two fundamental and interacting sources. First, synthesis and sequencing errors perturb the intended base mixture proportions. As the mixture complexity increases, this perturbation leads to higher inference error rates for composite letters at a given sequencing depth (Fig. 1c). Second, statistical sampling bias at low sequencing coverage leads to fluctuations in the observed base-frequency

distributions, making it difficult to accurately infer the original composition. Together, these factors reduce the discriminability of similar base mixtures, thereby increasing error rates and necessitating higher coverage to ensure reliable readout (Fig. 1d).

(2) Lines 54–61: The current summary of prior work on composite-letter DNA storage is incomplete and may inadvertently mislead readers about the actual performance trade-offs.

(i) Six-letter experiments (Σ_6). Anavy et al. (2019) demonstrated data recovery at $29\times$ coverage using KL-divergence inference followed by Fountain and Reed–Solomon coding. More recently, Xu et al. (2024) showed that a soft-decision decoder (Derrick-cp) applied to the same six-letter alphabet reduced the practical coverage threshold to $17\times$. Including this update will provide a realistic comparison point.

(ii) Large alphabets (256 letters). The early estimates of $2000\times$ coverage from Anavy et al. (2019)'s simulations with a 256-letter composite alphabet stemmed from high-density aims (~ 8 bits/letter). Xu et al. (2024)'s Derrick-cp soft-decision decoding algorithm reduced requirements to $490\times$. Importantly, these elevated coverage needs are not signs of irrational design, but rather inherent consequences of pursuing high-density information encoding. The key question is whether advanced strategies (KL-divergence, soft-decision, or the present set-partitioning) can offset this cost effectively.

Response: We thank the reviewer for highlighting these important developments. In response, we revised the relevant text in the manuscript (Lines 57–65) to incorporate the recent results by Xu *et al.* (2024).

We also clarified the context of large-alphabet encoding schemes, emphasizing that the expanded composite alphabets (e.g., 256-letter systems) are the valid approach to achieving higher information density. Furthermore, we stressed the critical role of robust letter detection and decoding algorithms, particularly those incorporating soft information, in effectively reducing coverage requirements and lowering sequencing costs. According to the suggestion above, we revised main text in the manuscript as follows.

Lines 57–65:

In prior work with a six-letter composite alphabet (Σ_6), Anavy et al.⁹ demonstrated data recovery at $29\times$ sequencing coverage using Kullback–Leibler (KL) divergence inference, Fountain coding, and Reed–Solomon (RS) coding^{26–28}. More recently, Xu et al.²⁹ applied a soft-decision decoding algorithm (Derrick-cp) to the same alphabet, reducing the practical coverage threshold to $17\times$. For larger composite alphabets, such as a 256-letter scheme targeting ultra-high logical density (theoretically up to 8 bits/letter), simulations by Anavy et al. showed that approximately $2000\times$ coverage would be required. The recent Derrick-cp algorithm reduced this requirement to $490\times$, highlighting the potential of advanced decoding methods²⁹.

(3) Lines 80-82: The current Fig. 1f and Extended Data Table 1 inaccurately report Xu et al.'s metrics: Their six-letter performance derived from *in vitro* experiments following Anavy et al., not simulations; The value of 2.06 bits/cycle includes primer and index regions, while Extended Data Table 1 consistently omits these overheads. Excluding primers/indexes, the correct density is 2.39 bits/cycle. The revised methods and corresponding numerical values are listed in bellowed table.

Therefore, in direct comparison with Xu et al.—considering both logical density and coverage—the performance of the current eight-letter system is comparable, but does not demonstrate notable advantages. I recommend updating Fig. 1f and Extended Data Table 1 to correct these values and revising the accompanying text to reflect that actual performance.

Response: We sincerely thank the reviewer for the detailed and constructive comment. We apologize for the inaccuracy in our previous reporting of Xu et al.'s work.

We have updated both **Fig. 1f** and **Supplementary Table 1** to reflect the accurate value for the six-letter system, and clarified that this metric is derived from *in vitro* experimental data. Correspondingly, we have revised the main text in the manuscript (**Lines 89–101**) to provide a more accurate and balanced comparison. During revision, we further conducted synthesis experiments with a 15-letter system, achieving a logical density of 3.125 bits per letter (excluding primers/indices) and error-free data recovery at 33× coverage.

According to the suggestions above, we revised main text in the manuscript as follows. The modified figures and tables are also listed below.

Lines 89–101:

Compared with the previous six-letter composite systems^{18,19}, our eight-letter storage system demonstrated advantages in logical density and physical redundancy. In comparison with state-of-the-art soft-decision inference algorithms, our system achieved comparable performance in terms of density and sequencing coverage (Fig. 1f and Supplementary Table 1). Specifically, Xu et al.²⁹ reported a logical density of 2.39 bits per letter (excluding primers/indices) at a coverage of 17× using a soft-decision decoder, while our system achieved 2.5 bits per letter at a similar coverage level. We further designed large-scale oligonucleotide pools containing 10,000 composite strands. Both the eight-letter and 15-letter alphabets were experimentally validated using array-based synthesis. For the 15-letter system, our composite letter detection method enabled error-free recovery at 33× sequencing coverage. The system achieved an information density of 3.125 bits per letter. Overall, the proposed 15-letter storage system demonstrated advantages in the number of composite letters, experimental synthesis scale, storage density, and coverage required for data recovery.

In the **main manuscript**, **Fig. 1f** is revised to update the comparison with prior work.

In the **Supplementary Information**, **Supplementary Table 1** is revised to provide a detailed comparison, including the coding scheme, total data volume, alphabet size,

logical density, and the sequencing coverage required for error-free recovery.

Figure 1f. Sequencing coverage and logical density of the present work compared with published composite alphabets.

Supplementary Table 1. Comparison of published composite letter data storage schemes.

Study	Error correction	Alphabet	Data volume	Total composite letters	Logical density (composite letters, bits/letter)	Average coverage	
Exp.	Anavy et al. ¹	RS(45, 43) +Fountain (1.1)	Σ_6	2.12 MB	7,830,000	2.16	29×
		RS(45, 43) +Fountain (1.08)	Σ_6	6.42 MB	23,490,000	2.18	29×
		RS(45, 43) +Fountain (1.08)	Σ_5	6.42 MB	26,055,000	1.97	29×
		Binary Huffman	Σ_{15}	19.25 Bytes	42	3.67	100×
		Binary Huffman	Σ_{20}	22.5 Bytes	42	4.29	100×
		Choi et al. ²	RS	Σ_{15}	854 Bytes	1,890	3.61
		RS	Σ_6	135 Bytes	499,833	2.16	250×
	Xu et al. ⁶	RS(45, 41) over $GF(7^3)$	Σ_6	6.42 MB	34,391,250	2.39	17×
	This work	NB-LDPC(3780, 1260) over $GF(2^6)$	Σ_8	945 Bytes*3	7,560*3	1	9×
		RS(1890, 1575) over $GF(2^{12})$	Σ_8	2,355 Bytes	7,560	2.5	14×
RS(3750, 3125) over $GF(2^{12})$		Σ_8	187,500 Bytes*2	600,000*2	2.5	16×	
RS(30000, 25000) over $GF(2^{15})$		Σ_{15}	234,375 Bytes*2	600,000*2	3.125	33×	
Sim.	Anavy et al. ¹	RS(136, 130)	Σ_{256}	2.12 MB	2,373,975	7.13	2000×
		RS(68, 65)	Σ_{56}	2.12 MB	3,354,480	5.05	500×
	Xu et al. ⁶	RS(45, 41) over $GF(2^6)$	Σ_{64}	20.8 MB	31,495,500	5.46	190×
		RS(45, 41) over $GF(2^7)$	Σ_{128}	20.8 MB	27,776,250	6.37	300×
		RS(45, 41) over $GF(2^8)$	Σ_{256}	20.8 MB	24,273,000	7.28	490×
	This work	RS(32767, 27767) over $GF(2^{15})$	Σ_{15}	780,713 Bytes	1,966,020	3.18	30×

2. The main motivation for composite alphabets is to reduce synthesis cycles, which dominate the costs in DNA Storage system. However, the current manuscript does not quantify this critical trade-off. A comparative cost analysis aligned with prior studies would clarify the practical benefits of the proposed scheme.

Response: We thank the reviewer for this valuable suggestion.

The costs in the DNA storage system associated with our eight-letter and 15-letter schemes were estimated by quantifying total synthesis and sequencing cost. We conducted the comparative cost analysis aligned with prior studies from two perspectives.

(1) Comparison of the total synthesis and sequencing costs. Synthesis costs are primarily determined by logical density, while sequencing cost depends on the required coverage for successful decoding.

We estimated the number of synthesized nucleotides needed to store 1 MB of data using the reported logical density of each method^{R1}. Sequencing cost was calculated as the product of coverage and nucleotide count^{R2}. For consistency, we assumed a sequencing cost of \$0.0000012/100 nt and a synthesis cost of \$0.05/100 nt. To account for index and amplification overhead, we considered both “payload-only” and “all synthesized nucleotides” cases (**Supplementary Fig. 68**). Compared with the standard four-base design of Organick et al.^{R3}, our eight- and 15-letter designs reduce synthesis cost by 56% and 64%, respectively. Compared with recent six-letter systems, the eight-letter design lowers synthesis cost by 13% (Anavy et al.^{R4}) and 4% (Xu et al.^{R5}), and the 15-letter design by 30% and 23%, respectively. Even after accounting for indexing and primer overheads, our design remains more cost-effective than the traditional four-base approach, with a slightly higher cost resulting from the use of a more robust double-end indices scheme. Detailed calculations are provided in **Supplementary Note 6**.

(2) Normalized cost under different synthesis-to-sequencing cost ratios. Under our cost model, the synthesis-to-sequencing cost ratio ($C_{\text{syn}} : C_{\text{seq}}$) can be as high as 40,000:1, though this ratio is expected to decline as synthesis technology advances. We further normalized total costs under different synthesis-to-sequencing cost ratios ($C_{\text{syn}} : C_{\text{seq}}$), using the standard four-base scheme of Organick et al.^{R3}. (requiring 5× sequencing depth) as the baseline. Representative ratios of 1000:1 and 500:1 were evaluated, and the costs of different schemes were scaled relative to the baseline work (**Supplementary Table 10**). Across all prior studies, composite-letter systems outperform the conventional four-base system. When considering only the composite-letter payload, the total cost of our system was reduced by 56% (eight-letter) and 64% (15-letter) at a 1000:1 ratio; at 500:1, the reductions remained 55% and 63%, respectively. Details are included in **Supplementary Table 10**.

In the **main manuscript**, Part 6 of the *Results* (“Cost analysis on composite letter DNA storage at low coverage”) is added to describe the synthesis of our composite letters and to provide an analysis of the associated costs.

In the **Supplementary information**:

(1) **Supplementary Table 10** is added to present a detailed comparison of normalized costs.

(2) **Supplementary Fig. 68** is added to illustrate the overall cost, including synthesis and sequencing, as well as a cost analysis reflecting the “write once, read many” characteristic of DNA storage.

(3) **Supplementary Note 6**, titled “Cost analysis on composite letter DNA storage at low coverage” is added to provide a detailed description of the cost calculation methods.

References

[R1]. Erlich, Y. & Zielinski, D. DNA Fountain enables a robust and efficient storage architecture. *Science* **355**, 950–954 (2017).

[R2]. Wetterstrand KA. DNA sequencing costs: data from the NHGRI genome sequencing program (GSP). Available at: www.genome.gov/sequencingcostsdata (2019).

[R3]. Organick, L. et al. Random access in large-scale DNA data storage. *Nat. Biotechnol.* **36**, 242–248 (2018).

[R4]. Anavy, L., Vaknin, I., Atar, O., Amit, R. & Yakhini, Z. Data storage in DNA with fewer synthesis cycles using composite DNA letters. *Nat. Biotechnol.* **37**, 1229–1236 (2019).

[R5]. Xu, Y., Ding, L., Wu, S. & Ruan, J. Overcoming the high error rate of composite DNA letters-based digital storage through soft-decision decoding. *Adv. Sci.* **11**, 2402951 (2024).

Supplementary Figure 68. Cost analysis of different DNA storage schemes. Estimated costs for writing (left) and reading (right) 1 MB of data, assuming DNA synthesis and sequencing costs of \$0.05/100 nt and \$0.0000012/100 nt, respectively. Two conditions are considered: payload sequences (light bars) and all synthesized nucleotides including primers and indices (dark bars).

Supplementary Table 10. Comparative cost analysis aligned with prior studies.

Study	Alphabet	Cov.	Logical density		Normalized cost			
			(bits/nt)		$C_{\text{syn}}:C_{\text{seq}}$ (500:1)		$C_{\text{syn}}:C_{\text{seq}}$ (1000:1)	
			Payload	Including primers + indices	Payload	Including primers + indices	Payload	Including primers + indices
Organick et al.¹⁵	Σ_4	5×	1.10	0.81	1.00	1.00	1.00	1.00
Goldman et al.¹⁶	Σ_4	51×	0.34	0.19	3.53	4.65	3.38	4.46
Erlich et al.³	Σ_4	10.5×	1.86	1.19	0.60	0.70	0.59	0.68
Anavy et al.¹	Σ_5	29×	1.97	1.37	0.58	0.62	0.57	0.61
	Σ_6	29×	2.18	1.52	0.53	0.56	0.52	0.55
	Σ_{15}	100×	3.67	1.56	0.36	0.62	0.33	0.57
	Σ_{20}	100×	4.29	1.82	0.30	0.53	0.28	0.49
Choi et al.²	Σ_6	250×	2.16	1.50	0.76	0.80	0.63	0.67
	Σ_{15}	250×	3.61	1.79	0.45	0.67	0.38	0.56
Xu et al.⁶	Σ_6	17×	2.39	2.06	0.47	0.40	0.47	0.40
This work	Σ_8	14×	2.50	1.29	0.45	0.64	0.44	0.63
	Σ_{15}	35×	3.125	1.67	0.37	0.51	0.36	0.50

3. Supplementary Tables 6–8 show that set-partitioning (SP) yields much lower error rates for standard bases (A/T/C/G) but higher error rates for composite letters (R, Y, M, K, etc.), which dominate the overall error budget, compared to KL-divergence (e.g. at 15×, error ≈4.8% for composites under SP vs 4.2% under KL). The authors should conduct end-to-end decoding performance comparison, using the same outer error-correction scheme (e.g., Reed-Solomon or LDPC), with detection via either set-partitioning or KL-divergence. This will clarify real gains (if any) in net data recovery. Furthermore, Xu et al. (2024) demonstrated that a MAP-based decoder can achieve >99% letter accuracy at 19× coverage—and still 98.3% accuracy even at 15×—clearly outperforming SP (demonstrated in Fig 5d and Supplementary Tables 6–8). The authors should therefore include MAP in their performance comparison to elucidate SP’s relative strengths and limitations.

Response: We thank the reviewer for the constructive comments. In response, we have incorporated the state-of-the-art maximum a posteriori (MAP) detection scheme into the performance analysis. We conducted end-to-end decoding comparisons of SP, KL and MAP, using the same outer error-correction codes.

(1) End-to-end validation with practical raw sequencing data.

Based on synthesis and sequencing experiments, we performed six sets of experimental comparisons, including two eight-letter designs with 126 composite strands each and four large-scale designs (two eight-letter and two 15-letter systems with 10,000 composite strands each). For SP, KL and MAP methods, using raw sequencing data, we evaluated both letter detection accuracy and end-to-end decoding performance under different average coverages (Supplementary Figs. 42–46).

(2) Letter detection accuracy and required coverage for error-free recovery

At low coverage, sampling bias leads to insufficient strand copies for some sequences. To mitigate this, we applied a copy threshold during data readout: sequences with fewer than four copies were treated as erasures and corrected by the outer code, thereby improved overall detection accuracy.

In the two small-scale pool experiments with eight-letter alphabets, one composite pool was encoded using a high-reliability LDPC code and the other using a high-rate RS code. In both experiments, the MAP method achieved 99% detection accuracy at 12× and 13× sequencing coverage, respectively, while our SP method reached approximately 98.9% accuracy at 13× coverage (Supplementary Table 7). When using the same RS code with a coding rate of 0.83, MAP achieved error-free recovery at 12× coverage, whereas SP and KL required 14× coverage (Supplementary Fig. 42). In contrast, when using a low-rate LDPC code ($R=1/3$), erasures became the dominant factor affecting recovery at low sequencing depths. Although SP and KL exhibited slightly lower detection accuracy than MAP, all three schemes achieved error-free recovery at 9× coverage (Supplementary Fig. 43).

In the two large-scale pool experiments with eight-letter alphabets, each pool

contained 10,000 composite strands synthesized via inkjet chemistry. When combined with an outer RS code ($R=0.83$), the results were consistent with the small-scale validation: the MAP algorithm achieved the highest detection accuracy and therefore required the lowest sequencing coverage ($18\times$ and $15\times$ for the two pools), followed by SP ($19\times$ and $16\times$, respectively) (Supplementary Figs. 44 and 45).

For the two 15-letter systems, MAP remained the most accurate method, reaching $>98\%$ detection accuracy at approximately $26\times$ coverage and achieving error-free recovery at $26\times$ and $27\times$ coverage (Supplementary Figs. 46 and 47; Supplementary Table 8). In contrast, SP suffered from partitioning errors near set boundaries, leading to slightly reduced accuracy (98%) and requiring $33\times$ and $35\times$ coverage for error-free recovery. The KL approach was more affected, requiring approximately $42\times$ and $48\times$ coverage. Furthermore, we established a simulation model to evaluate recovery performance for 32,767 composite strands under a simulated base error rate of 0.3% . Consistently, MAP required the lowest sequencing coverage, followed by SP and KL (Fig. 3d, e).

(3) Advantages and limitations of the proposed SP scheme

Unlike MAP, which directly compares posterior probabilities across all candidate letters, our set-partitioning (SP) approach reduces complexity by first grouping letters and then applying maximum likelihood estimation within each subset. This substantially narrows the candidate space and lowers computational cost. Across all six experiments, SP consistently demonstrated lower time complexity compared with MAP and KL, confirming its computational efficiency advantage (Supplementary Figs. 42–46).

We acknowledge that with larger alphabets, SP exhibited reduced detection accuracy relative to MAP due to boundary effects in set partitioning. In this contribution, we limited to 15-letter schemes.

We revised the manuscript as follows.

(3) Advantages and limitations of the proposed SP scheme

Unlike MAP, which directly compares posterior probabilities across all candidate letters, our set-partitioning (SP) approach reduces complexity by first grouping letters and then applying maximum likelihood estimation within each subset. This substantially narrows the candidate space and lowers computational cost. Across all six experiments, SP consistently demonstrated lower time complexity compared with MAP and KL, confirming its computational efficiency advantage.

We acknowledge that with larger alphabets, SP may exhibit reduced detection accuracy relative to MAP due to boundary effects in set partitioning. In this contribution, we limited to 15-letter schemes.

The revised manuscript as follows.

Lines 226–236:

Finally, we compared the end-to-end recovery performance of our set partitioning (SP) method with the KL method¹⁹ and the maximum a posteriori probability (MAP) method²⁹ with computer simulations (Fig. 3d, e). MAP performs global inference and shows the highest detection accuracy but the longest runtime. SP reduces the candidate space through partitioning, resulting in a slight degradation in accuracy, also a reduction in computational complexity. Both SP and MAP methods achieved lower error rates than the KL method. With 1,000 independent trials using 16 threads on the same server (Intel Xeon Gold 5220R CPU@2.20 GHz and 256 GB RAM), SP consumed only 25% of the time compared to the MAP method. In the simulation settings, MAP achieved error-free recovery at an average coverage of 25×, SP at 30×, whereas KL required 45× coverage. The performance in vitro experiments will be further verified in the following sections.

Lines 260–277:

Then, the minimum sequencing coverage for data recovery was evaluated with 1,000 independent trials on two sequencing datasets. Both the erasure error rate and the letter error rate progressively decreased as the average coverage increased (Fig. 4d, e). At low sequencing coverage, erasures dominated, while with increasing average coverage, erasures were suppressed, enabling reliable recovery. Using the NB-LDPC code with a code rate of 0.33, error-free recovery was achieved at an average coverage of 9×, even with an erasure error rate as high as 7% (Fig. 4f). Using RS code with a code rate of 0.83, at least 14× coverage was required for error-free recovery.

Third, we benchmarked SP method against the KL and MAP approaches at low sequencing coverages. In the column-based eight-letter experiments and the array-based experiments of 10,000 strands (including two eight-letter and two 15-letter systems), MAP consistently achieved the highest detection accuracy (Supplementary Tables 7 and 8). SP, as the second-best in accuracy, showed clear advantages in runtime over both MAP and KL (Supplementary Figs. 42 and 43). For the eight-letter pool of 10,000 strands, MAP enabled error-free recovery at 15× coverage, compared with 16× for SP. Executed with 10 threads, SP completed 300 validation runs in ~10 s (Supplementary Figs. 44 and 45). For the 15-letter system, MAP also provided the highest accuracy (Supplementary Figs. 46 and 47), while SP showed boundary-related precision loss yet still achieved error-free recovery at 33× coverage, consistent with simulations.

In the Supplementary information:

- (1) **Supplementary Table 7 and Table 8** are added to present the detailed error rates of each composite letter.
- (2) **Supplementary Figs. 42–47** are added to show the data recovery performance of the SP method and its comparison with the KL and MAP methods.

126 composite strands, 116 letters, Alphabet {A,T,G,C,R,Y,M,K}, RS(1890,1575)

Supplementary Figure 42. End-to-end decoding performance of the eight-letter system (126 strands, 116 letters, RS code, $R=5/6$). **a**, Relationship between raw sequencing coverage and valid base coverage used for data recovery. **b**, 94.86% of reads passed length filtering and 99.96% were successfully identified by double-end indices. **c**, 99% accuracy at 13 \times for MAP and 98.9% for SP. **d**, Erasure rate at different average coverages. **e**, Percentage of successful error-free recovery across 1,000 independent trials per coverage. All three methods achieved error-free recovery at 14 \times coverage. **f**, Runtime comparison of SP, KL, and MAP methods in single-thread mode.

126 composite strands, 116 letters, Alphabet {A,T,G,C,R,Y,M,K}, NB-LDPC(3780,1260)

Supplementary Figure 43. End-to-end decoding performance of the eight-letter system (126 strands, 116 letters, NB-LDPC code, $R=1/3$). **a**, Relationship between raw sequencing coverage and valid base coverage used for data recovery. **b**, 94.95% of reads passed length filtering, and 99.96% were successfully identified by indices. **c**, Letter error rate at different average coverages. **d**, Erasure errors were the major limiting factor for recovery, with the erasure rate exceeding 20% at 6 \times coverage and remaining as high as 5% at 10 \times . **e**, Percentage of successful recoveries across 1,000 independent trials. With the strong error-correction capability of the outer code ($R=1/3$), all three methods achieved successful recovery at 9 \times coverage. **f**, Runtime comparison in single-thread mode, where SP maintained a ~25% runtime advantage over MAP.

10,000 composite strands, 124 letters, Alphabet {A,T,G,C,R,Y,M,K}, RS(3750, 3125)

Supplementary Figure 44. End-to-end decoding performance of the eight-letter system (10,000 strands, 124 letters). **a**, Relationship between raw sequencing coverage and valid base coverage used for data recovery. **b**, 85.72% of reads passed length filtering and 99.82% were successfully identified by indices. **c**, Letter error rates show that SP closely matches MAP, with both outperforming KL in accuracy. **d**, At the coverage of 17 \times , the erasure rate was very low (<1%). **e**, Percentage of successful recoveries across 300 independent trials at different coverages. MAP achieved error-free recovery at 18 \times coverage, while SP, with a slight loss in accuracy, achieved error-free recovery at 19 \times . **f**, Runtime comparison of SP, KL, and MAP in multi-threaded mode (10 threads), values represent the cumulative runtime of 300 trials. SP requires less than 15 s, compared with 20 s for KL and 25 s for MAP.

10,000 composite strands, 112 letters, Alphabet {A,T,G,C,R,Y,M,K}, RS(3750, 3125)

Supplementary Figure 45. End-to-end decoding performance of the eight-letter system (10,000 strands, 112 letters). **a**, Relationship between raw sequencing coverage and valid base coverage used for data recovery. **b**, 87.53% of reads passed length filtering and 99.94% were successfully identified by indices. **c**, Detection accuracy was comparable among the three methods, with MAP performing the best. **d**, Erasure rates decreased below 0.5% by 16× coverage. **e**, Error-free recovery was achieved by MAP at 15×, and by SP and KL at 16× coverage, based on 300 independent trials. **f**, SP completed 300 trials in ~12 s, whereas MAP required ~25 s.

Supplementary Figure 46. End-to-end decoding performance of the 15-letter system (10,000 strands, 124 letters). **a**, Relationship between raw sequencing coverage and valid base coverage used for data recovery. **b**, 86.38% of reads passed length filtering and 99.93% were successfully identified by indices. **c**, With larger alphabets, SP showed higher error rates than MAP. **d**, At the coverage $\geq 25\times$, erasure errors were nearly absent. **e**, Percentage of successful recoveries across 300 independent trials at each coverage. Error-free recovery was achieved at $26\times$ coverage using MAP, whereas $33\times$ coverage was required for SP. **f**, Runtime comparison for 300 trials: SP completed letter detection in ~ 18 s, compared with ~ 35 s for the KL method and >60 s for the MAP method.

Supplementary Figure 47. End-to-end decoding performance of the 15-letter system (10,000 strands, 112 letters). **a**, Relationship between raw sequencing coverage and valid base coverage used for data recovery. **b**, 86.66% of reads passed length filtering and 99.96% were successfully identified by indices. **c**, With larger alphabets, SP showed higher error rates than MAP. **d**, At the coverage $\geq 26\times$, erasure errors were nearly absent. **e**, Percentage of successful recoveries across 300 independent trials at each coverage. Error-free recovery was achieved at $27\times$ coverage using MAP, whereas $35\times$ coverage was required using SP and $48\times$ with the KL method. **f**, Runtime comparison for 300 trials: SP completed letter detection in ~ 20 s, compared with ~ 35 s for the KL method and ~ 60 s for the MAP method.

Supplementary Table 7. Letter detection error rates under different detection methods with an eight-letter alphabet {A, T, G, C, R, Y, M, K} and RS(1890, 1575).

Method	Cov.	A(%)	T(%)	G(%)	C(%)	R(%)	Y(%)	M(%)	K(%)	Total
SP	10×	0.129	0.089	0.138	0.093	0.294	0.454	0.293	0.264	1.754
	11×	0.123	0.082	0.126	0.087	0.243	0.390	0.246	0.223	1.519
	12×	0.117	0.075	0.116	0.081	0.204	0.341	0.206	0.187	1.327
	13×	0.113	0.070	0.108	0.076	0.172	0.298	0.175	0.159	1.170
	14×	0.110	0.066	0.100	0.072	0.145	0.262	0.150	0.137	1.042
	15×	0.105	0.061	0.093	0.068	0.122	0.229	0.131	0.117	0.926
	16×	0.103	0.058	0.089	0.065	0.104	0.200	0.112	0.104	0.834
	17×	0.101	0.055	0.085	0.064	0.090	0.176	0.097	0.091	0.758
	18×	0.098	0.052	0.081	0.061	0.077	0.156	0.085	0.082	0.691
	19×	0.096	0.050	0.079	0.060	0.067	0.138	0.074	0.073	0.636
	20×	0.096	0.049	0.078	0.059	0.060	0.123	0.066	0.066	0.597
21×	0.095	0.047	0.076	0.059	0.054	0.109	0.058	0.060	0.558	
KL	10×	0.215	0.153	0.276	0.174	0.235	0.339	0.227	0.207	1.825
	11×	0.222	0.160	0.289	0.183	0.182	0.268	0.177	0.164	1.644
	12×	0.225	0.164	0.296	0.186	0.142	0.213	0.139	0.130	1.493
	13×	0.225	0.165	0.298	0.186	0.113	0.170	0.111	0.104	1.372
	14×	0.219	0.163	0.295	0.185	0.091	0.139	0.090	0.086	1.268
	15×	0.213	0.159	0.287	0.183	0.073	0.114	0.074	0.072	1.175
	16×	0.206	0.154	0.279	0.178	0.060	0.094	0.062	0.063	1.096
	17×	0.197	0.148	0.267	0.172	0.051	0.080	0.052	0.056	1.022
	18×	0.188	0.141	0.255	0.166	0.045	0.068	0.046	0.051	0.959
	19×	0.183	0.136	0.243	0.161	0.039	0.059	0.040	0.046	0.907
	20×	0.178	0.131	0.232	0.155	0.035	0.054	0.036	0.044	0.864
21×	0.173	0.126	0.220	0.149	0.032	0.047	0.033	0.041	0.820	
MAP	10×	0.126	0.088	0.161	0.108	0.250	0.377	0.243	0.219	1.572
	11×	0.116	0.082	0.149	0.102	0.200	0.312	0.196	0.179	1.336
	12×	0.109	0.076	0.139	0.096	0.162	0.261	0.161	0.146	1.150
	13×	0.102	0.071	0.127	0.090	0.135	0.222	0.135	0.121	1.002
	14×	0.098	0.066	0.117	0.086	0.113	0.193	0.116	0.104	0.892
	15×	0.094	0.062	0.108	0.082	0.096	0.169	0.102	0.090	0.801
	16×	0.092	0.060	0.102	0.078	0.083	0.148	0.089	0.082	0.735
	17×	0.089	0.058	0.097	0.076	0.074	0.132	0.081	0.075	0.680
	18×	0.086	0.053	0.090	0.072	0.067	0.119	0.074	0.070	0.631
	19×	0.084	0.051	0.086	0.071	0.061	0.108	0.069	0.065	0.594
	20×	0.082	0.049	0.083	0.069	0.058	0.101	0.066	0.063	0.572
21×	0.082	0.048	0.081	0.068	0.055	0.092	0.063	0.061	0.550	

Supplementary Table 8. Letter detection error rates under different detection methods with a 15-letter alphabet (10,000 composite strands, 124 letters).

Method	Cov.	A(%)	T(%)	G(%)	C(%)	R(%)	Y(%)	M(%)	K(%)	S(%)	W(%)	H(%)	B(%)	V(%)	D(%)	N(%)	Total
SP	26×	0.037	0.044	0.073	0.032	0.245	0.129	0.109	0.225	0.240	0.151	0.272	0.375	0.184	0.381	0.809	3.305
	30×	0.033	0.041	0.063	0.029	0.203	0.098	0.088	0.171	0.198	0.114	0.198	0.285	0.156	0.291	0.633	2.602
	34×	0.030	0.040	0.056	0.027	0.167	0.078	0.072	0.135	0.160	0.090	0.148	0.220	0.140	0.228	0.502	2.090
	38×	0.028	0.038	0.051	0.025	0.141	0.066	0.061	0.110	0.132	0.074	0.113	0.173	0.127	0.181	0.402	1.722
	42×	0.026	0.037	0.048	0.024	0.126	0.059	0.054	0.094	0.116	0.064	0.088	0.137	0.115	0.146	0.324	1.457
	46×	0.024	0.037	0.045	0.023	0.119	0.054	0.051	0.084	0.106	0.058	0.070	0.110	0.103	0.119	0.263	1.265
	50×	0.023	0.036	0.042	0.022	0.114	0.051	0.049	0.077	0.101	0.054	0.056	0.089	0.090	0.098	0.215	1.119
KL	26×	0.174	0.166	0.415	0.146	0.461	0.155	0.196	0.324	0.447	0.228	0.111	0.137	0.192	0.185	0.266	3.431
	30×	0.145	0.138	0.325	0.120	0.493	0.159	0.208	0.336	0.479	0.236	0.087	0.108	0.197	0.164	0.155	3.349
	34×	0.128	0.122	0.263	0.104	0.490	0.155	0.205	0.330	0.477	0.230	0.079	0.097	0.211	0.160	0.091	3.141
	38×	0.119	0.112	0.225	0.095	0.453	0.144	0.189	0.306	0.441	0.212	0.077	0.093	0.226	0.162	0.055	2.909
	42×	0.113	0.108	0.202	0.089	0.396	0.129	0.164	0.270	0.384	0.187	0.077	0.092	0.237	0.164	0.034	2.647
	46×	0.108	0.104	0.187	0.085	0.335	0.114	0.139	0.234	0.323	0.161	0.076	0.091	0.238	0.164	0.021	2.379
	50×	0.105	0.101	0.174	0.082	0.284	0.101	0.118	0.203	0.269	0.141	0.073	0.087	0.230	0.158	0.014	2.138
MAP	26×	0.040	0.046	0.067	0.035	0.108	0.102	0.050	0.182	0.104	0.115	0.197	0.251	0.145	0.264	0.448	2.152
	30×	0.036	0.043	0.058	0.031	0.079	0.074	0.036	0.132	0.075	0.082	0.138	0.182	0.097	0.192	0.356	1.611
	34×	0.033	0.040	0.051	0.029	0.064	0.060	0.029	0.103	0.059	0.064	0.097	0.131	0.065	0.140	0.278	1.243
	38×	0.030	0.039	0.046	0.027	0.054	0.051	0.025	0.086	0.050	0.054	0.069	0.097	0.045	0.105	0.210	0.990
	42×	0.028	0.037	0.043	0.026	0.048	0.046	0.023	0.075	0.045	0.047	0.052	0.074	0.033	0.082	0.156	0.815
	46×	0.027	0.037	0.040	0.025	0.044	0.044	0.021	0.068	0.041	0.043	0.040	0.058	0.025	0.066	0.116	0.694
	50×	0.026	0.036	0.038	0.024	0.041	0.042	0.020	0.063	0.039	0.040	0.033	0.047	0.021	0.055	0.087	0.611

4. While the conceptual use of set-partitioning is sound, its current entropy-based implementation appears suboptimal for composite letters. The results suggest the proposed method reduces within-subset confusion at the expense of mis-grouping between subsets. The authors appear to have directly ported the telecommunications partitioning approach without adaptation to composite letter characteristics. I suggest to develop a more tailored partitioning approach to accurately separate subsets.

Response: We appreciate the reviewer's comment. Set partitioning is a general method in telecommunication signal design^[R1-R3]. Our approach is inspired by the concept of set partitioning in telecommunications. We extended the partitioning approach using discrete entropy values according to the characteristics of composite letters.

First, our proposed method is based on the discrete entropy values of the different letters. Specifically, our design is grounded in the DNA diamond model, which organizes the 15 composite letters into spatial positions—vertices, edges, face centers, and centroid—within a tetrahedral geometry (**Figure 1e**). This geometric arrangement naturally reflects discrete entropy values (0, 1.0, 1.58, 2.0) (**Supplementary Fig. 7**). In the practical synthesis and sequencing, the entropy distribution was broadened yet retained discriminability, indicating that the method remains effective (**Supplementary Fig. 26**). Then, we proposed an entropy-based set-partitioning strategy. In the proposed scheme, a two-stage method was used. During the second stage, maximum a posteriori (MAP) was executed on a smaller subset. Therefore, the decoding complexity was greatly reduced.

In contrast, most of the methods for communications^{R1}, including our scheme for indel correction^{R2, R3}, were based on a distance metric, e.g., Euclidean distance. As illustrated by the classical example of eight Phase-Shift Keying (8PSK) modulation (**Figure R2**), eight symbols are evenly spaced on the unit circle at 45° intervals, representing three bits per symbol. Set-partitioning is a hierarchical technique that groups symbols into subsets with progressively increasing minimum Euclidean distance, thereby enabling multilevel decoding with reduced bit error rates. After the first decomposition, the adjacent points are evenly spaced on the unit circle at 90° intervals. The distance between the points is enlarged. After the second decomposition, the adjacent points are evenly spaced on the unit circle at 180° intervals. The distances between the points are largest for 8PSK signal.

Second, with our signal design, the synthesis of composite letter is very convenient and compatible with the existing synthesizers. Users only needed to add the monomers in the equal proportion. Recently, there were some other design guidelines proposed in information theory⁴⁰⁻⁴³. These schemes may obtain better error correction performance by accurately designing the base probabilities of each letter. However, they were not convenient for synthesis, for each letter requires a very accurate control of four different monomers.

We acknowledge that entropy-based partitioning may introduce some cross-subset misclassification. It can be predicted from a small overlap of the different distribution of signal in different subsets. However, its primary benefit lies in the reduction of

computational complexity and synthesis complexity.

References

[R1]. Yazdani R, Ardakani M. Reliable communication over non-binary insertion/deletion channels[J]. *IEEE Transactions on Communications*, 2012, 60(12): 3597-3608.

[R2]. Liu Y, Chen W. Iterative Decoding for the Concatenated Code to Correct Nonbinary Insertions/Deletions[C]//2017 *IEEE 85th Vehicular Technology Conference (VTC Spring)*. IEEE, 2017: 1-5.

[R3]. Liu Y, Chen W. Decoding on adaptively pruned trellis for correcting synchronization errors[J]. *China Communications*, 2017, 14(7): 1-9.

Figure R2. The decoding process for 8PSK using the set-partitioning method in conventional telecommunication systems.

Supplementary Figure 7. a, Information entropy values of the 15 letters used in this study, exhibiting only four discrete entropy values.

Supplementary Figure 26. Entropy distribution under practical synthesis and sequencing. **a and b**, Entropy distributions of the 15-letter alphabet at high coverage (1,000×) and the minimum coverage for error-free decoding (33×). **c and d**, Entropy distributions of the eight-letter alphabet at high coverage (1,000×) and the minimum coverage for error-free decoding (19×). Each curve represents the entropy distribution for individual composite letters under noisy sequencing conditions.

5. The manuscript reports discarding ~23% of reads during length filtering (Fig. 5g), a shocking large fraction for Illumina sequencing (which typically has very low indel rates). The authors should explain the source of these indels and verify that filtering did not bias the results. Additionally, it is unclear whether the stated coverage thresholds (e.g. 18× for error-free recovery and in Fig. 5e-f) refer to pre- or post-filter read counts. Since coverage directly translates to DNA-storage readout cost, it should be defined on the basis of raw sequencing depth rather than the subset surviving filtering.

Response: We thank the reviewer for this insightful comment. We updated the computation of sequencing coverage. We updated all the results with raw sequencing data. We thank the reviewer for pointing out this point. In this situation, we observed the ratios of the filtered reads reduced. Reads removed by length filtering primarily arose from induced insertion/deletion errors in the synthesis step, which caused deviations from the designed payload length.

(1) On the read filtering ratio of length filtering

We have updated the data. In our prior version of manuscript, we used clean data with quality control (QC). The QC may induce bias on the reads. Therefore, we filter about 23% reads during length filtering. We thank the reviewer. According to the comments, we have updated the results on raw data. The filtered ratio is about 5%.

However, it is still quite high considering the relatively low indel errors of Illumina sequencing. We conjectured that indels may stem from the synthesis process. For the column-based synthesis, the indel error rate was low, and the filtered ratio was also low. For the low-cost synthesis using inkjet printing, the indel rate was relatively high, and the filtered ratio was higher (**Supplementary Figs. 56 and 57**). The filtered ratios were quite identical to the indel error ratio. Correspondingly, the filtered ratios increased to greater than 10% (**Supplementary Table 9**). For the low-cost synthesis using inkjet printing, low-cost protocols were adopted for cost and no rigorous purification was performed. Thus, the indel especially deletion rate was higher.

Unfortunately, unlike conventional DNA storage schemes, composite letter DNA storage currently lacks effective strategies for handling indels. We incorporated a length filtering step into our data recovery workflow (**Supplementary Fig. 58**) to prevent such errors from interfering composite letter detection.

We down-sampled the raw reads to target coverages (e.g., 10×, 12×, 14×), and then ran the complete workflow with and without length filtering. For each coverage, 1,000 independent random trials were conducted. Results from both coding schemes (coding rate of 5/6 and 1/3) consistently demonstrated that removing off-length reads improved base detection accuracy (**Fig. 5e, f and Supplementary Fig. 67**). For instance, at 14× coverage, length filtering reduced the composite-letter error rate from an average of 2.8% to 1.0% (accuracy > 99%), and when combined with RS decoding ($R=5/6$), achieved error-free recovery in 1000/1000 trials. In contrast, without length filtering, only ~30% of trials achieved error-free recovery under the same (pre-filter) coverage condition.

(2) On the sequencing coverage

In the prior version of our manuscript, we used clean data. According to the reviewer's kind comments, we have modified all the results with the raw sequencing data. In this new version, the coverage values reported in this study are all defined based on the raw sequencing depth after paired-end assembly, not on the subset of filtered reads.

We updated **Part 5 of the Results** with a revised performance analysis of length filtering. The revised text in the manuscript is as follow.

Lines 328–337:

Under the same average coverage, we evaluated the effectiveness of length filtering in reducing the symbol error rate by comparing two schemes. Across both coding schemes ($R=0.83$ and $R=0.33$), filtering consistently improved letter detection accuracy (Fig. 5e, f and Supplementary Fig. 67). At the $14\times$ coverage, filtering reduced the average composite-letter error rate from 2.8% to 1.0% (accuracy > 99%). With RS code ($R=0.83$), length filtering enabled error-free recovery in all trials. Without filtering, only ~30% of trials achieved error-free recovery under the same coverage. Although filtering decreased the number of usable reads, 94.39% were retained after primer identification, length filtering, and index identification, ensuring high-quality input for downstream detection (Fig. 5g). Supplementary Table 9 presents the results of all pools, including small-scale pools and large-scale pools.

Supplementary Table 9. Length filtering of the different experiments.

	Exp.1	Exp.2	Exp.3	Exp.4	Exp.5	Exp.6	
Composite strands	126	126	10,000	10,000	10,000	10,000	
Alphabet	8	8	8	8	15	15	
Length (Primer+index)	28 nt	28 nt	32 nt	26 nt	32 nt	26 nt	
Error rate (Forward primer+index)	Total	0.0059	0.0062	0.0102	0.0030	0.0087	0.0134
	Ins.	0.0001	0.0001	0.0004	0.0001	0.0004	0.0001
	Del.	0.0053	0.0056	0.0085	0.0027	0.0071	0.0129
	Sub.	0.0005	0.0005	0.0013	0.0002	0.0012	0.0004
Error rate (Reverse primer+index)	Total	0.0020	0.0020	0.0040	0.0009	0.0036	0.0036
	Ins.	0.0001	0.0001	0.0003	0.0001	0.0003	0.0001
	Del.	0.0013	0.0013	0.0025	0.0005	0.0021	0.0031
	Sub.	0.0006	0.0006	0.0012	0.0003	0.0012	0.0004
Length (Index+payload)	76	76	84	72	84	72	
Length filtering (Valid reads)	94.9%	94.9%	85.9%	87.6%	86.5%	86.7%	
Length filtering (Invalid reads)	5.1%	5.1%	14.1%	12.4%	13.5%	13.3%	

Supplementary Figure 56. Length distribution of assembled sequencing reads. a and b, Length distributions of 126 composite strands. The strands were encoded using either the NB-LDPC(3780, 1260) code or the RS(1890, 1575) code, with 79.6% and 78.5% of sequencing reads, respectively, reaching the designed full length of 116 nt. **c and d**, Large-scale composite pools using the eight-letter alphabet, where 56.0% and 73.4% of reads, respectively, reached the designed full length after paired-end assembly. **e and f**, Large-scale composite pools using the fifteen-letter alphabet, where 63.0% and 55.9% of reads, respectively, reached the designed full length after paired-end assembly.

Supplementary Figure 57. Length distributions of index and payload regions after primer identification. **a and b**, Both datasets exhibited nearly identical distributions, with 94.9% of reads retaining the designed full length (76 nt) after primer trimming. **c and d**, Two large-scale pools using the eight-letter alphabet, with 85.9% (84 nt) and 87.6% (72 nt) of reads, respectively, retaining the designed payload length. **e and f**, Two large-scale pools using the fifteen-letter alphabet, with 86.5% (84 nt) and 86.7% (72 nt) of reads, respectively, retaining the designed payload length.

Supplementary Figure 58. Readout workflow with length filtering. Sequencing reads are first aligned to double-end primers to identify payload and index segments (Primer alignment). Reads with abnormal length are discarded, ensuring equal-length, aligned sequences for downstream readout (Length filtering). Reads are then grouped into different clusters according to the index sequences (Index identification). Within each cluster, base frequency distributions at each position are calculated across multiple copies, and the most probable composite letter is inferred (Letter detection).

Fig. 5 | Performance evaluation considering interference and indels. Panels a–d are shown in the main text and omitted here for clarity. **e**, The length filtering step prevents error propagation and improves letter detection accuracy. Results are based on 1,000 independent trials at each coverage, with error bars indicating the standard deviation. **f**, The length filtering step reduces the sequencing coverage required for data recovery. Plots show the number of successful recoveries out of 1,000 independent trials at each coverage. **g**, The proportion of effective reads after primer identification, length filtering, and index identification.

126 composite strands*116 letters, NB-LDPC(3780,1260), Alphabet {A,T,G,C,R,Y,M,K}

Supplementary Figure 67. Letter detection accuracy and recovery performance with length filtering. **a**, Letter error rates as a function of average coverage, comparing workflows with (red) and without (blue) length filtering. Length filtering effectively reduces the error rate across all coverages. **b**, Successful recovery rate as a function of average coverage. For each coverage, 1,000 independent evaluations were performed, and the successful recovery rate represents the proportion of tests achieving error-free recovery. Although the omission of length filtering leads to higher error rates, error-free recovery can still be achieved using NB-LDPC coding with a rate of 1/3. **c**, Proportion of valid reads at each processing step of the recovery workflow. Starting from 6,066,534 raw paired-end assembled reads (100%), 99.23% were retained after primer identification, 94.18% after length filtering, and 94.15% after index identification.

6. The manuscript lacks critical detail regarding the synthesis methodology for composite letters, which is a key practical challenge in composite DNA storage. In particular, it is not clear whether each oligo encoding a composite letter is individually designed (deterministic assignment of A/T/C/G at each oligo) or whether the synthesis process uses probabilistic base mixing at degenerate positions, resulting in random incorporation across molecules. This distinction impacts scalability, and decoding accuracy. Also, discuss scalability implications for large-scale oligo pools.

Response:

We thank the reviewer for raising this valuable comment. We quite agree with the reviewer. The specific synthesis of composite letters is indeed a key practical aspect of composite DNA storage. In all our design, we did not individually design each strand. During the chemical synthesis process, we used mixed monomers at each degenerate positions.

We also agree with the reviewer that there may be some concerns on the scalability and decoding accuracy. We performed several practical verifications on 10,000 composite strands using eight-letter and 15-letter alphabets.

We provide detailed clarification below.

1. Synthesis methodology for composite letters

In our work, each composite DNA strand was synthesized independently in a dedicated well using column-based or array-based phosphoramidite chemistry.

For column-based synthesis, during each coupling cycle, composite letters were generated by mixing different bases, i.e., preprogrammed volumetric blending of A, T, C, G monomers at specified ratios (**Supplementary Fig. 4a and Supplementary Table 2**). This approach implements composite letters through controlled degeneracy rather than by designing deterministic single-base sequences.

Moreover, in our scheme, we used eight or 15 letters. Each composite letter was indeed mixed at the equal ratio (**Supplementary Table 2**). For column-based synthesis, we only needed to pre-mix two or three or four monomers with the equal molecule quantities. It is quite convenient. The synthesizer can only support eight inputs for monomers. Therefore, we used this synthesizer platform to support **Alphabets 1–3**.

For inkjet array-based synthesis, the mixture process was implemented by controlling the equal jetting volumes of different monomers according to the specific composite letter within one coupling cycle. Therefore, this platform can at least support 15 different composite letters. We tested the **Alphabets 1 and 4**. That is, Alphabet 1, {A, T, G, C, R, Y, M, K} and Alphabet 4, {A, T, G, C, R, M, Y, K, S, W, H, B, V, D, N}.

2. Practical column-based synthesis and data recovery.

We performed four synthesis experiments based on eight different composite-letter alphabets. In each experiment, 126 composite strands were independently synthesized across 126 CPG columns and subsequently stored in 126 individual tubes. These composite strands were synthesized by Hippobio Co., Ltd. An eight-channel oligo synthesizer was used, which can precisely control volumetric delivery of each channel (Supplementary Fig. 4b). Eight channels delivered the pure or mixed monomers, corresponding to eight composite letters. Successful data recovery was achieved directly on raw sequencing reads on the pooled oligos (Supplementary Figs. 42 and 43).

3. Scalability implications for large-scale oligo pools

According to the kind comments, we performed four oligonucleotide pool experiments, each synthesizing 10,000 composite strands. The strand lengths were 112 letters and 124 letters, respectively. The length of composite letter length in each strand was 60. Two different alphabets (Alphabets 1 and 4, eight-letter and 15-letter) were employed (Supplementary Fig. 4c). The other parts were primers and indices with the length of 60nt using standard four bases.

These pools were synthesized by Dynegene Technologies, using array-based inkjet oligonucleotide synthesis. The pools were sequenced using NGS in PE150 mode. Successful data recovery was achieved directly on raw reads (Supplementary Figs. 44–47).

Therefore, we expect that the scalability for large-scale oligo pools is promising, even for larger synthesized pools.

We added **Part 6 of Results Section** to include a description of the experimental scheme.

Lines 176–185:

For practical writing into DNA, two synthesis platforms were used to test different composite alphabet implementations (Supplementary Fig. 4). First, 126-strand pools were constructed with eight-letter alphabets using column-based phosphoramidite synthesis. Each strand was synthesized in an individual well, with composite letters implemented by volumetric mixing of A, T, C, and G phosphoramidite monomers. Second, large-scale pools containing 10,000 composite strands were synthesized by array-based inkjet phosphoramidite chemistry. In all experiments, payloads were uniformly 60 letters, flanked with double-end indices and PCR primer sequences (Fig. 2e). For the column-based pools of 126 strands, each carried two indices (8 nt) and two primers (20 nt) at the 5' and 3' ends. For the array-based pools of 10,000 strands, the indices were 6 nt or 12 nt at the two ends, respectively.

We updated the **Methods section** to include a description of the synthesis of composite-letter oligonucleotide pools at different scales.

Lines 473–481:

Synthesis of small-scale pools (column-based synthesis)

A total of four small-scale composite-letter oligonucleotide pools were prepared. For each pool, 126 composite DNA sequences were synthesized using Oligo-192 Oligo Synthesizer by Hippobio (Huzhou, China). Each strand was synthesized independently in a dedicated column, with composite letters implemented through volumetric mixing of A, T, G, and C phosphoramidite monomers. HPLC Purification was performed. All 126 samples were then manually combined in equimolar amounts to generate one master pool. Four such master pools were obtained, with final DNA concentrations of 2,482 ng/ μ L, 2,515 ng/ μ L, 301 ng/ μ L, and 459 ng/ μ L.

Lines 483–491:

Synthesis of large-scale pools (array-based inkjet printing synthesis)

Four large-scale oligonucleotide pools were designed, each comprising 10,000 composite strands: two based on the eight-letter alphabet and two based on the 15-letter alphabet. These pools were synthesized by Dynegene Technologies (Shanghai, China) using array-based inkjet printing synthesizer. During synthesis, four inkjet nozzles dispensed monomer droplets onto array chips according to predefined base-mixing ratios. The eight-letter system used the alphabet {A, T, G, C, R, Y, M, K}, and the 15-letter system used the alphabet {A, T, G, C, R, M, Y, K, S, W, H, B, V, D, N}. Each large-scale pool was delivered as a lyophilized master pool containing 3 μ g of DNA.

Supplementary Figure 4. Illustration of the composite strand synthesis with column-based and array-based synthesizers. **a**, Column-based synthesis of composite DNA strands. Eight reagent channels correspond to eight composite bases. Three composite alphabets were chosen and 126 composite strands were synthesized for each alphabet. **b**, Each designed composite strand, composed of composite letters (e.g., B, G, T, D, A, etc.), is individually synthesized in a dedicated well using column-based DNA synthesis. **c**, Array-based inkjet synthesis of large-scale composite-letter pools. Four printheads deliver monomer droplets onto array chips following predefined base mixing ratios.

Minor reversions

Line 141: The Galois field definition for NB-LDPC code appears incorrect—please correct the GF notation.

Response: Thank you for your valuable comments. We are sorry for this error and have corrected this error. The manuscript has been revised as follows:

Lines 156–158:

For data encoding, in the eight-letter system, 945 bytes of text were first encoded using NB-LDPC(3780,1260) code ($R=0.33$) over Galois Field ($GF(64)$), with interleaving to mitigate burst erasures³⁹.

Response to the Reviewers' Comments

We sincerely thank both reviewers for their valuable comments on our manuscript and greatly appreciate their constructive remarks and suggestions. We have revised the manuscript accordingly based on their valuable feedback and provide detailed point-by-point responses below. All changes in the revised manuscript are highlighted in yellow.

Response to the Reviewers #1

I appreciate the authors for their detailed responses, which have addressed most of my earlier concerns. However, one remaining issue relates to the statement in the revised manuscript that “homopolymer length distributions in composite systems are comparable to conventional four-letter storage.” In fact, conventional DNA data storage systems do not generally use unconstrained four-letter encoding. A widely adopted strategy (Goldman et al., Nature, 2013) is to convert the binary data to base-3 digits (trits), and then convert to DNA code by replacement of each trit with one of the three nucleotides different from the previous one used. This strategy explicitly prevents homopolymers in every DNA strand. Thus, the authors may clarify that conventional systems can avoid homopolymers through appropriate encoding algorithms rather than simply employing 4-letter representations. More details can be found in this paper:

Goldman, N., Bertone, P., Chen, S. et al. Towards practical, high-capacity, low-maintenance information storage in synthesized DNA. Nature 494, 77–80 (2013).

Response: We thank the reviewer for pointing out this problem. We completely agree with the reviewer. We have revised this point. We clarified that our comparison refers specifically to four-letter storage without homopolymer optimization. Several existing DNA storage systems avoided homopolymers through appropriate encoding algorithms, e.g., rotation coding or homopolymer filtering.

The revised main text in the manuscript is as follows:

Lines 294–302:

Homopolymer length distributions in composite systems are similar to four-letter DNA storage without homopolymer optimization or random sequences (Supplementary Fig. 55). Several DNA storage systems guaranteed homopolymer length through appropriate encoding, e.g., rotation coding⁴ or homopolymer filtering⁹. We did not use any optimization on homopolymer, balancing the efficiency and performance⁴⁰. To address these challenges, we incorporated two tailored strategies into the readout pipeline: robust double-end indices to mitigate strand crosstalk and DNA degradation; length filtering to reduce the impact of indel errors during synthesis and sequencing.

Response to the Reviewers #2

I thank the authors for their considerable efforts in revising this manuscript. The manuscript is strengthened by the inclusion of 15-letter in vitro experiments, data written through two composite-letter synthesis platforms tailored to different oligo scales, and additional degradation experiments assessing DNA stability. With Illumina sequencing for readout, all decoding experiments achieved 100% data recovery, demonstrating the robustness of the proposed framework. Overall, the concept of “DNA diamond” is original and appealing, offering a useful entropy-based perspective on composite-letter codec for DNA storage.

For further clarification and fairness in reporting, I have a few minor suggestions:

1. When comparing overall performance (lines 86–100), please pair sequencing coverage with a net information density that includes all synthesized nucleotides (including primers and indices), or at least add the index overhead. In this design, the double-ended indices are intentionally introduced to mitigate strand displacement/oligo breaks and degradation (lines 301–302) and therefore form part of the error-correction mechanism that enables lower recovery depth; omitting them can overstate payload efficiency. By contrast, in prior studies such as Xu et al. and Anavy et al., indices primarily serve addressing, so payload-only metric would be reasonable within those contexts. The cost analysis should follow the consistent accounting and incorporate index costs to ensure fair comparison.

Response: We thank the reviewer for the kind suggestion. Following this advice, we have recalculated the logical densities explicitly including index and payload. Our encoded indices have two functions. First, they served as reliable addresses avoiding strand crosstalk. Reliable read addressing is important to achieve low recovery coverage. Second, they can identify degraded oligos due to break. The revised values are now reported in **Supplementary Table 1** and **Supplementary Table 4**. We also revised the cost analysis considering the index cost, and accordingly updated **Supplementary Table 10** and **Supplementary Figure 68**.

The revised main text in the manuscript is as follows:

Lines 85–97:

Specifically, we implemented the eight-letter alphabet using column-based synthesis. Error-free data recovery was achieved at 14× sequencing coverage. This scheme increased the information density from the canonical 2 bits per letter to 2.5 bits per letter (excluding primers and indices). **The logical density of payload and indices was 1.97 bits per letter.** In comparison with state-of-the-art schemes, our system achieved comparable performance in terms of density and sequencing coverage (Fig. 1f and Supplementary Table 1). We further designed large-scale oligonucleotide pools containing 10,000 composite strands. Both the eight-letter and 15-letter alphabets were experimentally validated using array-based synthesis. For the 15-letter system, our

composite letter detection method enabled error-free recovery at 33× sequencing coverage. The system achieved a logical density of 3.125 bits per letter for payload, corresponding to a density of 1.51 bits per letter for payload and indices. Our work validated the feasibility of using a 15-letter alphabet for DNA data storage (10,000 strands synthesized in a single pool) and demonstrated that error-free recovery can be achieved at relatively low sequencing coverage.

In the **Supplementary information**:

- (1) **Supplementary Table 1** was updated to include the logical density that considering index overhead;
- (2) **Supplementary Table 4** was revised to incorporate the logical density that considering index overhead;
- (3) **Supplementary Table 10** was revised to present the cost analysis considering index overhead;
- (4) **Supplementary Fig. 68** was also revised to incorporate the synthesis and sequencing cost calculations that account for index overhead.

Supplementary Table 1. Comparison of published composite letter data storage schemes.

Study	Error correction codes	Alphabet	Data volume	Total composite letters	Logical density (composite letters, bits/letter)	Logical density (including indices)	Average coverage
Anavy et al. ¹	RS(45, 43) +Fountain (1.1)	Σ_6	2.12 MB	7,830,000	2.16	1.93	29×
	RS(45, 43) +Fountain (1.08)	Σ_6	6.42 MB	23,490,000	2.19	1.96	29×
	RS(45, 43) +Fountain (1.08)	Σ_5	6.42 MB	26,055,000	1.97	1.76	29×
	Binary Huffman	Σ_{15}	19.25 Bytes	42	3.67	3.67 (only one strand, no index)	100×
	Binary Huffman	Σ_{20}	22.5 Bytes	42	4.29	4.29 (only one strand, no index)	100×
	Exp. Choi et al. ²	RS	Σ_{15}	854 Bytes	1,890	3.61	3.37
RS		Σ_6	135 Bytes	499,833	2.16	2.0	250×
Xu et al. ⁶	RS(45, 41) over $GF(7^3)$	Σ_6	6.42 MB	34,391,250	2.39	2.06	17×
This work	NB-LDPC(3780, 1260) over $GF(2^6)$	Σ_8	945 Bytes*3	7,560*3	1	0.79	9×
	RS(1890, 1575) over $GF(2^{12})$	Σ_8	2,355 Bytes	7,560	2.5	1.97	14×
	RS(3750, 3125) over $GF(2^{12})$	Σ_8	187,500 Bytes*2	600,000*2	2.5	2.08 (112 letters) / 1.79 (124 letters)	16× / 19×
	RS(30000, 25000) over $GF(2^{15})$	Σ_{15}	234,375 Bytes*2	600,000*2	3.125	2.60 (112 letters) / 2.23 (124 letters)	35× / 33×
Anavy et al. ¹	RS(136, 130)	Σ_{256}	2.12 MB	2,373,975	7.13	/	2000×
	RS(68, 65)	Σ_{56}	2.12 MB	3,354,480	5.05	/	500×
Sim. Xu et al. ⁶	RS(45, 41) over $GF(2^6)$	Σ_{64}	20.8 MB	31,495,500	5.46	/	190×
	RS(45, 41) over $GF(2^7)$	Σ_{128}	20.8 MB	27,776,250	6.37	/	300×
	RS(45, 41) over $GF(2^8)$	Σ_{256}	20.8 MB	24,273,000	7.28	/	490×
This work	RS(32767, 27767) over $GF(2^{15})$	Σ_{15}	780,713 Bytes	1,966,020	3.18	/	30×

Supplementary Table 4. Coding potential and the net information density.

Scheme	Column-based DNA synthesis experiments		Array-based DNA synthesis experiments				Simulation
Data volume (bytes)	945	2,355	187,500	234,375	234,375	780,713	
Bit information (bits)	7,560	18,900	1,500,000	1,875,000	1,875,000	6,247,575	
Payload	126*60	126*60	10000*60	10000*60	10000*60	32767*60	
($M*L$)	=7560 letters	=7560 letters	=600,000 letters	=600,000 letters	=600,000 letters	=1,966,020 letters	
Alphabet	Σ_8	Σ_8	Σ_8	Σ_{15}	Σ_{15}	Σ_{15}	
Coding potential (bits/letter)	$\log_2(8) = 3$	$\log_2(8) = 3$	$\log_2(8) = 3$	$\log_2(15) = 3.9$	$\log_2(15) = 3.9$	$\log_2(15) = 3.9$	
Coding scheme	NB-LDPC(3780, 1260)	RS(1890, 1575)	RS(3750, 3125)	RS(30000, 25000)	RS(30000, 25000)	RS(32767, 27767)	
Code rate	$R=1/3$	$R=5/6$	$R=5/6$	$R=5/6$	$R=5/6$	$R=17/20$	
Logical density (payload)	1 bit/letter	2.5 bits/letter	2.5 bits/letter	3.125 bits/letter	3.125 bits/letter	3.18 bits/letter	
Synthesized nucleotides (including primers/indices)	126*116 =14,616	126*116 =14,616	10000*124 =1,240,000	10000*112 =1,120,000	10000*124 =1,240,000	10000*112 =1,120,000	/
Logical density (payload+index)	0.79 bit/letter	1.97 bits/letter	1.79 bits/letter	2.08 bits/letter	2.23 bits/letter	2.60 bits/letter	/
Net information density (synthesized nucleotides) (bits/letter)	0.52 bits/letter	1.29 bits/letter	1.21 bits/letter	1.34 bits/letter	1.51 bits/letter	1.67 bits/letter	/

Supplementary Table 10. Comparative cost analysis aligned with prior studies.

Alphabet	Study	Cov.	Logical density (bits/letter) (excluding primers)	Normalized cost	
				$C_{\text{syn}}:C_{\text{seq}}$ (500:1)	$C_{\text{syn}}:C_{\text{seq}}$ (1000:1)
Σ_4	Organick et al.¹⁵	5×	1.10	1.00	1.00
	Goldman et al.¹⁶	51×	0.29	4.14	3.97
	Erlich et al.³	10.5×	1.55	0.72	0.71
Σ_5	Anavy et al.¹	29×	1.76	0.65	0.64
Σ_6	Anavy et al.¹	29×	1.96	0.59	0.57
	Choi et al.²	250×	2.0	0.82	0.68
	Xu et al.⁶	17×	2.06	0.55	0.54
Σ_8	This work	16×	2.08	0.54	0.53
Σ_{15}	Anavy et al.¹	100×	3.67	0.36	0.33
	Choi et al.²	250×	3.37	0.48	0.41
	This work	35×	2.60	0.45	0.44
Σ_{20}	Anavy et al.¹	100×	4.29	0.30	0.28

Supplementary Figure 68. Cost analysis of different DNA storage schemes. Estimated costs for writing (left) and reading (right) 1 MB of data, assuming DNA synthesis and sequencing costs of \$0.05/100 nt and \$0.0000012/100 nt, respectively. Two conditions are considered: Only payload part (light bars) and the region including payload and indices (dark bars).

2. When positioning against prior studies (lines 351–354), please compare like-for-like alphabet sizes (e.g., 15-letter vs earlier 15-letter systems; for 8-letter vs 6-letter.).

Response: We thank the reviewer for this valuable suggestion. We have revised the synthesis cost comparison. We modified the comparison in Supplementary Fig. 68 and Supplementary Table 10.

The revised text in the main manuscript is as follows:

Lines 348–355:

On the one hand, only considering payload part, our eight-letter and 15-letter systems reduced the synthesis cost by 47% and 57%, respectively, compared with the conventional four-base DNA storage system²⁵. On the other hand, considering the index overhead (excluding primers), for the similar alphabet size of six or eight, we obtained the similar cost reductions. For the 15-letter alphabet, we achieved large-scale validation using 10,000 strands also with a slight increase on synthesis cost (Supplementary Fig. 68). This was primarily due to the adoption of a more robust double-end index scheme that introduced additional redundancy to improve reliability.

Lines 358–361:

Only considering the region excluding primers at a 1000:1 ratio, the eight-letter and 15-letter systems reduced the total storage cost by 47% and 56%, respectively. When the ratio was reduced to 500:1, the total cost reductions remained at 46% and 55%, respectively.

3. The statements near lines 99–101 about advantages of the 15-letter system should be calibrated to match the evidence shown in Fig. 1f and Supplementary Table 10, where—under comparable 15-letter conditions—the storage density appears lower than that in Anavy et al.

Response: We thank the reviewer for pointing out this problem. Our scheme indeed did not show superiority in storage density over that of Anavy et al., for we used more overhead for large-scale verification. In order to support a practical large-scale verification experiment (10,000 strands in a single pool), more overhead was allocated to indices and reliable error-correcting codes with more redundancy were adopted for the payload part. We advanced in experiment scale for composite DNA storage with 15-letter alphabet.

The revised text in the main manuscript is as follows:

Lines: 95–97:

Our work validated the feasibility of using a 15-letter alphabet for DNA data storage (10,000 strands synthesized in a single pool) and demonstrated that error-free recovery can be achieved at relatively low sequencing coverage.

Response to the Reviewers' Comments

We sincerely thank the reviewer for the valuable comments and constructive suggestions provided during the review of our manuscript. The manuscript has been carefully revised according to the reviewer's minor suggestions, and the main text has been further improved. All changes in the revised manuscript are highlighted in yellow.

Response to the Reviewers #2

I appreciate the authors' detailed response and the rigorous revisions made to the manuscript, which have adequately addressed my previous concerns. Before publication, I have just two minor suggestions to harmonize the text with the supplementary data.

1. Lines 93 – 97: The revised text reports a density of 1.51 bits/letter for "payload and indices", which appears to reference the net information density (including primers). According to Supplementary Tables 1 and 4, the correct logical density for "payload and indices" is 2.23 bits/letter. Please correct the main text to align with your tabular data.

Response: We thank the reviewer for carefully checking and kind comment. We are very sorry for this obvious mistake. When considering "Payload and Indices" together, the logical density was 2.23 bits/letter. We misused the logical density of "Payload + Indices + Primers", which was 1.51 bits/letter.

In the revised manuscript, the statement has been corrected so that the logical density for "payload and indices" is now reported as 2.23 bits/letter, in agreement with Supplementary Tables 1 and 4.

Lines 95–96:

The system achieved a logical density of 3.125 bits per letter for payload, corresponding to a density of 2.23 bits per letter for payload and indices.

2. Lines 22–25 and 379–381: When highlighting the system's overall performance, pairing the sequencing coverage with the logical density of "payload and indices" (e.g., 33× sequencing coverage with 2.23 bits/letter) is more appropriate. This ensures the reader fully appreciates the efficiency of the error-correction mechanism (the indices) that enables such low recovery depth.

Response: We thank the reviewer for this constructive suggestion. In the revised manuscript, the overall system performance described in both the Abstract (Lines 24–26) and the Discussion (Lines 379–381) has been updated to pair the sequencing coverage with the logical density of "payload and indices". In Table S4, we honestly

listed all the three logical densities in detail: Payload only (3.125 bits/letter); Payload + Indices (2.23 bits/letter); Payload + indices + Primers (1.51 bits/letter).

The revised main text in the manuscript is as follows:

Lines 24–26:

The full 15-letter constellation enables 3.125 bits per letter for payload with error-free recovery at 33× coverage, corresponding to a density of 2.23 bits per letter for payload plus indices.

Lines 381–383:

Notably, the 15-letter alphabet was experimentally realized at the 10,000-strand scale, achieving 3.125 bits per letter for payload (or 2.23 bits per letter for payload plus indices) and error-free recovery at 33× coverage.